

# Diurnal evolution of non-precipitating marine stratocumuli in an LES ensemble

Yao-Sheng Chen[1,2], Jianhao Zhang[1,2], Fabian Hoffmann[3], Takanobu Yamaguchi[1,2], Franziska Glassmeier[4], Xiaoli Zhou[1,2], and Graham Feingold[2]

[1]Cooperative Institute for Research in Environmental Sciences, University of Colorado Boulder, Boulder, Colorado, USA
[2]NOAA Chemical Sciences Laboratory, Boulder, Colorado, USA
[3]Meteorologisches Institut, Ludwig-Maximilians-Universität München, Munich, Germany
[4]Delft University of Technology, Delft, Netherlands

**Correspondence:** Yao-Sheng Chen (yaosheng.chen@noaa.gov)

**Abstract.** We explore the impacts of the diurnal cycle, free-tropospheric (FT) humidity values, and interactive surface fluxes on the cloud system evolution of non-precipitating marine stratocumuli based on a large ensemble of large-eddy simulations. Cases are separated into three categories based on their degree of decoupling and cloud liquid water path ($LWP_c$). A new budget analysis method is proposed to analyze the evolution of $LWP_c$ under both coupled and decoupled conditions. More

coupled clouds start with relatively low $LWP_c$ and cloud fraction ($f_c$) but experience the least decrease in $LWP_c$ and $f_c$ during the daytime. More decoupled clouds undergo greater daytime reduction in $LWP_c$ and $f_c$, especially those with higher $LWP_c$ at sunrise because they suffer from faster weakening of a net radiative cooling. During the nighttime, a positive correlation between FT humidity and $LWP_c$ emerges, consistent with higher FT humidity reducing both radiative cooling and the humidity jump, both of which reduce entrainment and increase $LWP_c$. The time rate of change in the $LWP_c$ is more likely to be negative

for higher $LWP_c$ and greater inversion base height ($z_i$), conditions under which entrainment dominates as turbulence develops. In the morning, the rate of the $LWP_c$ reduction depends on the $LWP_c$ at sunrise, $z_i$, and the degree of decoupling, with distinct contributions from subsidence and radiation. Under well-mixed conditions, it takes about 10 h for the surface fluxes to offset 15% of the changes in entrainment warming and drying, assuming no changes in transfer coefficients or surface wind speed.

## 1   Introduction

Subtropical marine stratocumuli cover vast areas of Earth's surface and play an important role in Earth's energy balance by reflecting solar radiation back to space. A cloud reflects more solar radiation when its liquid water is distributed amongst a larger number of aerosol particles to form more numerous and smaller cloud droplets (Twomey, 1974, 1977). This initial effect propagates to other cloud properties through a series of complex processes, e.g., suppression of precipitation formation (Albrecht, 1989; Pincus and Baker, 1994), enhancement of cloud-top entrainment (Bretherton et al., 2007; Wang et al., 2003), and

an increase in solar absorption (Boers and Mitchell, 1994). These processes, all considered part of aerosol–cloud interactions (ACIs), may offset one another and their importance depends on the cloud's properties, its environment, and the time scale of interest (Stevens and Feingold, 2009).





From observations alone, it is difficult to identify and quantify the details of the aforementioned processes (e.g., Gryspeerdt et al., 2019; Wall et al., 2023), given the incomplete information of observed clouds and their environments, including co-varying meteorology and aerosols, and often in the form of snapshots rather than temporal evolution of the same cloud field (Stevens and Feingold, 2009; Mülmenstüdt and Feingold, 2018). Despite recent efforts in inferring processes after constraining such co-variations (e.g., Zhang et al., 2022; Zhang and Feingold, 2023) and in quantifying the temporal evolution in the cloud responses to aerosol perturbations (e.g., Qiu et al., 2024; Smalley et al., 2024; Gryspeerdt et al., 2022), causality or process attribution remains a challenge. While opportunistic experiments, such as ship tracks, provide a way to observe the adjustment of cloud properties to additional aerosol, they are often limited in their ability to represent the wide range of conditions the marine stratocumuli reside in (e.g., Manshausen et al., 2022; Yuan et al., 2023; Toll et al., 2019).

Meanwhile, fine-scale numerical modeling has been used to provide process-level understanding of ACIs. Early work focused primarily on case studies with aerosol perturbation experiments (Sandu et al., 2008; Caldwell and Bretherton, 2009; Wang and Feingold, 2009; Wang et al., 2010; Chen et al., 2011; Yamaguchi et al., 2015; Possner et al., 2018; Kazil et al., 2021; Prabhakaran et al., 2023; Chun et al., 2023). Although much has been learned from these studies, they do not cover the wide range of real-world conditions.

Recent work by Feingold et al. (2016) and Glassmeier et al. (2019) took a different approach: exploring ACIs in large-eddy simulation (LES) ensembles of marine stratocumuli. They performed LESs of a large number of cases, each set up with different initial conditions specified by meteorological factors and aerosol number concentration. Instead of performing aerosol perturbation experiments for each combination of meteorological factors, they used experiment design techniques to optimize the sampling of the initial condition space and later distilled the information regarding ACIs from both the individual and collective behaviors of ensemble members.

This approach has proved to be fruitful. Based on an LES ensemble of more than 150 nocturnal marine stratocumulus simulations, Glassmeier et al. (2019) found that several cloud properties (cloud fraction, cloud albedo, and relative cloud radiative effect) of ensemble members can be well described in the state-space of liquid water path (LWP) and cloud droplet number concentration ($N_d$). Using the same LES ensemble, Hoffmann et al. (2020) showed that all non-precipitating cases in this ensemble approach a steady state LWP band from different parts of the state space: clouds starting with high LWP thin over time and clouds starting with low LWP, and possibly partial cloudiness, thicken over time. The authors further performed a budget analysis based on mixed-layer theory (MLT; Lilly, 1968) and demonstrated how the balance between radiative cooling, cloud-top entrainment warming and drying, and other processes shaped the $N_d$-dependence of steady state LWP. Glassmeier et al. (2021) estimated the magnitude and time scale of the LWP adjustment to an $N_d$ perturbation from the collective behavior of the ensemble members and used them to infer biases in using ship-track to estimate the climatological forcing of anthropogenic aerosol. Hoffmann et al. (2023) explored the evolution of precpitating and non-precipitating stratocumuli in the space of albedo and cloud fraction using another ensemble of 127 cases.

The environmental conditions covered in the LES ensembles used by these works can be expanded. For instance, the free-troposphere (FT) in these simulations was fairly dry, while in reality a moister FT reduces cloud-top radiative cooling and modulates cloud-top entrainment warming and drying (Ackerman et al., 2004; Eastman and Wood, 2018). In addition, the





surface fluxes in those simulations were either constants prescribed following DYCOMS-II RF02 (Ackerman et al., 2009) or interactive but only responding to local wind fluctuations with calm mean winds, leading to relatively weak surface fluxes.

Lastly, despite the insights gained from nocturnal simulations, the daytime behavior of marine stratocumulus population needs to be explored to understand the shortwave radiative effects of these clouds, which are more relevant to aerosol–cloud climate forcing and issues like marine cloud brightening (Latham, 1990; Feingold et al., 2024).

In this study, we explore the impacts of diurnal cycles, FT humidity values, and interactive surface fluxes on the cloud system evolution. The rest of this manuscript is organized as follows. We first introduce the model and simulation configurations in

Section 2 and then provide an overview of the LES ensemble in Section 3. Next, we introduce a new budget analysis method and present results in Section 4. With this method, we examine the nighttime and daytime evolution of individual cases in Section 5. A few specific issues will be discussed in Section 6, after which we end the paper with a summary in Section 7.

## 2 Model and simulations

All LESs for this study are performed using the System for Atmospheric Modeling (SAM; Khairoutdinov and Randall, 2003),

version 6.10.10. SAM solves the anelastic Navier-Stokes equations in finite difference representation for the atmosphere on the Arakawa C grid. Similar to recent work by Yamaguchi et al. (2017) and Glassmeier et al. (2019), SAM is configured with a fifth-order advection scheme by Yamaguchi et al. (2011) and Euler time integration scheme for scalars, a second-order center advection scheme and with the third-order Adams-Bashforth time integration scheme for momentum, a 1.5-order TKE-based subgrid model similar to Deardorff (1980), a bin-emulating bulk two-moment microphysics parameterization (Feingold

et al., 1998) assuming a log-normal aerosol size distribution with fixed size and width parameters, and the Rapid Radiative Transfer Model (RRTMG; Mlawer et al., 1997; Iacono et al., 2008) that is modified to take into account background profiles of temperature and moisture above the model domain top (Yamaguchi et al., 2015), which is critical for radiative transfer in shallow domain simulations.

Different from Yamaguchi et al. (2017) and Glassmeier et al. (2019), the SAM used for this work uses the total water mixing

ratio (sum of vapor and hydrometeors) and the total number concentration (sum of aerosol and drop number concentrations) as prognostic variables to ensure better closure of the budgets associated with these two quantities for advection and several other physical processes. As a result, the water vapor mixing ratio is diagnosed from the total water and hydrometeor mixing ratios and the aerosol number concentration is diagnosed from the total, cloud droplet, and rain drop number concentrations. See details in the last paragraph of Section 2 in Yamaguchi et al. (2019).

As in Feingold et al. (2016) and Glassmeier et al. (2019), the LES ensemble members are generated from perturbed initial conditions. The initial profiles of liquid water potential temperature ($\theta_l$) and total water mixing ratio ($q_t$) are each constructed from two parts: a well-mixed boundary layer (BL) profile including a sharp jump at the top of the BL and a FT profile based on ERA5 climatology (Hersbach et al., 2020) and the Marine ARM GPCI Investigation of Clouds (MAGIC) campaign (Lewis et al., 2012; Zhou et al., 2015) observations. The initial BL $\theta_l$ and $q_t$ profiles are controlled by five parameters: $\theta_l$ and $q_t$ in the

BL and their jumps, $\Delta\theta_l$ and $\Delta q_t$, across the inversion base at the height of $h_{mix}$. See Appendix A for details on the FT profiles





and the construction of the complete profiles. The initial aerosol number mixing ratio, specified by a sixth parameter, $N_a$, is uniform throughout the domain. The initial horizontal wind speed is 0 m s$^{-1}$ everywhere.

Hundreds of initial profiles are set up from sets of these six parameters randomly and independently drawn from their ranges: BL $\theta_l$ is drawn from 284 to 294 K, BL $q_t$ from 6.5 to 10.5 g kg$^{-1}$, $\Delta\theta_l$ from 6 to 10 K, $\Delta q_t$ from $-10$ to 0 g kg$^{-1}$, $h_{mix}$ from 500 to 1300 m, and $N_a$ from 30 to 500 mg$^{-1}$. Compared with the parameter ranges used in Glassmeier et al. (2019), the range for $\Delta q_t$ now covers $-6$ to 0 g kg$^{-1}$ to include conditions with more humid FT. All initial profiles with (1) height of lifted condensation level ($z_{LCL}$) between around 225 m and 1075 m, (2) a saturated layer (i.e., $h_{mix} > z_{LCL}$), and (3) FT $\theta_l$ and $q_t$ profiles falling between the minimum and maximum of the ERA5 climatological profiles are simulated with the lower boundary conditions and large-scale forcings described below, which are the same for all simulations.

First, the surface fluxes of sensible heat, latent heat, and momentum are computed based on Monin-Obukhov similarity. The sea surface temperature (SST) is fixed for all simulations at 292.4 K. Since the mean horizontal wind speed is close to 0 m s$^{-1}$ in the lowest model level as a result of the simulation setup, a constant horizontal wind speed of 7 m s$^{-1}$ is added to the surface local wind fluctuation when calculating sensible and latent heat fluxes to obtain realistic flux values. Both this wind speed and the SST are based on the ERA5 climatology from the same region and time period as described in Appendix A. Second, a constant surface aerosol flux of 70 cm$^{-2}$ s$^{-1}$, based on estimates by Kazil et al. (2011), is prescribed to offset the loss of aerosols through coalescence scavenging (Wang et al., 2010). Lastly, a time-invariant subsidence profile is imposed as

$$w_s = \begin{cases} -Dz, \ z < 2000 \text{ m} \\ -0.0075 \text{ m s}^{-1}, \ z \geq 2000 \text{ m}, \end{cases} \tag{1}$$

where the divergence $D = 3.75 \times 10^{-6}$ s$^{-1}$. No other large-scale forcing is present in the simulations.

The simulation domain is $48 \times 48 \times 2.5$ km$^3$ in the x-, y-, and z-dimensions with 200-m horizontal and 10-m vertical grid spacings. It uses periodic lateral boundary conditions and has a damping layer from 2 km to domain top. All simulations are initialized at 18:40 local time (LT; 03:00Z) and then advanced for 24 h with a 1-s time step. Sunrise occurs between 05:23 and 05:24 LT and sunset occurs between 18:36 and 18:37 LT.

For this study, we focus on non-precipitating cases, defined by a cloud-base precipitation rate of less than 0.5 mm day$^{-1}$ (Wood, 2012). We further exclude simulations with multi-layer clouds, including surface fog. Finally, we discard simulations where the cloud top ever reaches 2 km, the lower bound of the damping layer, to avoid unrealistic results. This leaves 245 cases for further investigation. The first 2-h of each simulation is excluded as the spin-up.

## 3 Overview of LES ensemble behavior

In this section, we present an overview of the evolution of the 245 non-precipitating cases in our LES ensemble. Following Glassmeier et al. (2019), we start with the trajectories in the plane of cloud droplet number concentration ($N_d$) and cloud liquid water path (LWP$_c$), both based on columns with cloud optical depth greater than 1, the definition of "cloudy column" in this work (Figure 1). During the nighttime, the cases that start with low LWP$_c$ experience an increase in LWP$_c$, while the behavior





of the high $LWP_c$ cases is not immediately clear. The nighttime cloud fractions ($f_c$) are usually high. At sunrise, 68% of cases have $f_c > 0.99$ and 86% cases have $f_c > 0.95$. During the daytime, all cases start to lose $LWP_c$ and $f_c$ right after sunrise or in the early morning. Between noon and 15:00, about 86% cases reach their lowest daytime $LWP_c$. In the last hour of the simulation, 94% cases are gaining $LWP_c$. Very low $f_c$ occurs for many cases in the afternoon.

### 3.1 Categorization of cases

To provide a more consolidated view of the evolution, we categorize the cases by their degree of decoupling in the morning because the diurnal decoupling (Nicholls, 1984; Turton and Nicholls, 1987) is a common feature of the cloud-topped marine BL diurnal cycle and we expect different diurnal cycles between more coupled and more decoupled cases. We compute the relative decoupling index (denoted with $\mathcal{D}$) defined by Kazil et al. (2017),

$$\mathcal{D} = \frac{\overline{z_{cb}} - \overline{z_{LCL}}}{\overline{z_{LCL}}}, \tag{2}$$

where $\overline{z_{cb}}$ and $\overline{z_{LCL}}$ are the mean cloud base height and mean lifting condensation level (LCL, determined from conditions in the lowest model level), both averaged for cloudy columns. This index is a variant of the subcloud decoupling index, $\overline{z_{cb}} - \overline{z_{LCL}}$, originally proposed by Jones et al. (2011). A small value of $\mathcal{D}$ is more likely to be coupled while a large value of $\mathcal{D}$ is more decoupled.

Figure 2a shows $\mathcal{D}$ at 09:40 LT in the plane of $LWP_c$ and domain-mean inversion base height ($z_i$, based on levels with the greatest vertical gradient of liquid water static energy in individual columns) at sunrise. Clouds with greater $\mathcal{D}$ tend to occur in deeper BLs; many of these clouds experience very low daytime $f_c$ minima (Figure 2b) unless they start with very high $LWP_c$ at sunrise, although most cases have daytime $f_c$ maxima that are close to overcast (not shown). Based on this finding, we divide the cases into three categories based on $\mathcal{D}$ at 09:40 and $LWP_c$ at sunrise (05:22): (1) lo$\mathcal{D}$loL ($\mathcal{D} \leq 1$), (2) hi$\mathcal{D}$loL ($\mathcal{D} > 1$ and $LWP_c \leq 180$ g m$^{-2}$, the highest $LWP_c$ for the lo$\mathcal{D}$loL category), and (3) hi$\mathcal{D}$hiL ($\mathcal{D} > 1$ and $LWP_c > 180$ g m$^{-2}$) for further analysis (Figure 2c). Figure 2d shows the time series of $\mathcal{D}$ by category. During the nighttime, the medians of $\mathcal{D}$ for all three categories are relatively small, suggesting more coupled conditions. Some cases in the hi$\mathcal{D}$loL and hi$\mathcal{D}$hiL categories always exhibit a higher degree of decoupling during the night. During the daytime, $\mathcal{D}$ for all three categories increases into the afternoon. Overall, cases in the lo$\mathcal{D}$loL category experience weaker decoupling with their $\mathcal{D}$ start to increase at a slower rate from a later time, compared with other two categories. Figure 2e shows the time series of median $\overline{z_{cb}}$ and median $\overline{z_{LCL}}$ by category. During the daytime, the median $\overline{z_{LCL}}$ decreases for both hi$\mathcal{D}$loL and hi$\mathcal{D}$hiL, consistent with a strengthening decoupling limiting the surface based mixed layer. This does not happen to lo$\mathcal{D}$loL. Also, both hi$\mathcal{D}$loL and hi$\mathcal{D}$hiL categories experience dramatic diurnal changes in median $\overline{z_{cb}}$ and the cloud depth, approximated with $z_i - \overline{z_{cb}}$. Even though the categorization is based on $\mathcal{D}$, it nicely separates the lo$\mathcal{D}$loL category from the other two categories through the daytime.

### 3.2 Cloud evolution by category

Figures 3a and 3b display the average time series of $LWP_c$ and $f_c$ for three categories. Among the three categories, the lo$\mathcal{D}$loL category shows the lowest nighttime $LWP_c$ and $f_c$. However, this category also has the smallest decrease in $LWP_c$ and $f_c$





during the day. By contrast, the hi$\mathcal{D}$loL category has greater LWP$_c$ and nearly overcast conditions ($f_c > 0.99$) at sunrise but

experiences a much more dramatic decrease in both LWP$_c$ and $f_c$. The hi$\mathcal{D}$hiL category has the highest LWP$_c$ and $f_c$ at sunrise among all three categories. This category also shows diurnal fluctuations of large amplitude in both LWP$_c$ and $f_c$ with the daytime minimum between the lo$\mathcal{D}$loL and hi$\mathcal{D}$loL categories for both variables. It reaches its lowest LWP$_c$ and lowest $f_c$ latest in the day among all three categories. At the end of the simulation, all three categories experience a recovery of both LWP$_c$ and $f_c$. At this stage, they all have similar LWP$_c$, indicating that the diurnal cycle imposes a strong constraint to narrow

the range of LWP$_c$. In constrast, the $f_c$ differs significantly: the lo$\mathcal{D}$loL category has the highest $f_c$ and the hi$\mathcal{D}$loL category the lowest $f_c$.

There is hysteresis in the mean trajectories of the three categories in the plane of $f_c$ and the cloud depth ($z_i - z_{cb}$) plane (Figure 3c). The trajectory of the lo$\mathcal{D}$loL category makes the smallest loop, which can be interpreted as the least diurnal variation in cloud aspect ratio (the ratio between the cloud depth and $f_c$). Clouds in the hi$\mathcal{D}$loL and hi$\mathcal{D}$hiL categories experience greater

variation in the aspect ratio, more so for the hi$\mathcal{D}$loL categories. We examine the 3-D cloud fields for selected cases from these two categories and find that clouds in both categories evolve into a cumulus-rising-into-stratocumulus structure by noon (not shown). The cloud bases of the cumuli lower slightly while the stratocumuli continue to thin and lose $f_c$. This transition lowers $\overline{z_{cb}}$ and leads to the segments in the trajectories where $f_c$ decreases but cloud depth starts to recover. As the clouds develop towards sunset, they regain $f_c$ to become stratiform again.

## 3.3 Surface fluxes

To end this overview, we examine the surface fluxes in the simulations (Figures 4a and 4b). At the end of the first 2-h of the simulations, both the ranges of surface sensible heat flux (SHF) and latent heat flux (LHF) from all simulations encompass the values prescribed in the DYCOMS-II RF02 case (i.e., 16 and 93 W m$^{-2}$, respectively). Afterwards, the SHF decreases over time until late afternoon as the SHF effectively brings the BL air temperature towards the SST (Figures 4a). The SHF is

175 the strongest in the lo$\mathcal{D}$loL category, followed by the hi$\mathcal{D}$hiL and then the hi$\mathcal{D}$loL categories. This is because the shallower BLs in our ensemble also tend to be colder due to the criteria applied in the initial profiles. (For example, for a shallow BL to be initially saturated, its $z_{LCL}$ needs to be lower, which is more likely when the initial BL $\theta_l$ is low. See more in Section 2.) LHF shows a smaller relative change throughout the day (Figures 4b). During the nighttime, the LHF for the lo$\mathcal{D}$loL category remains quite steady and that for the hi$\mathcal{D}$loL category even increases as the turbulence spins up. The LHF is also the strongest

in the lo$\mathcal{D}$loL category, while the LHF from the other two categories are comparable at all times.

Following Eq. 1 in Lilly (1968), the domain-mean surface sensible and latent heat fluxes (SHF and LHF) can be written as

$$\text{SHF} = C_T U (\theta_{\text{SST}} - \theta_{\text{air}}), \text{LHF} = C_q U (q_{\text{sat}}(\text{SST}) - q_{\text{v,air}}), \tag{3}$$

where the wind speed used for surface flux calculations ($U$), lowest model level air temperature and water vapor mixing ratio ($\theta_{\text{air}}$ and $q_{\text{v,air}}$) are also the domain-means. Recall that in our simulations, the SST is 292.4 K and equivalent to a potential

temperature, $\theta_{\text{SST}}$, of 290.9 K given the surface pressure used in the simulations. (See Appendix.) The saturation mixing ratio at SST ($q_{\text{sat}}(\text{SST})$) is approximately constant due to the negligible drift of surface pressure. Comparing Figures 4c–f with





Figures 4a–b, it is clear that the evolutions of the SHF and LHF in our simulations are driven primarily by $(\theta_{\mathrm{SST}} - \theta_{\mathrm{air}})$ and $(q_{\mathrm{sat}}(\mathrm{SST}) - q_{\mathrm{v,air}})$, respectively. On average, the transfer coefficients for SHF $(C_T)$ and for LHF $(C_q)$ that are diagnosed from Eq. (3) decrease slightly over time, although cases with $\theta_{\mathrm{air}}$ very close to $\theta_{\mathrm{SST}}$ see larger fluctuations in $C_T$. $U$ mostly ranges between 7 than 7.3 m s$^{-1}$ throughout the day (Figure S1) because they result from the summation of relatively weak local wind velocities and a large constant wind speed (7 m s$^{-1}$, see Section 2). Our results are consistent with the findings reported by Kazil et al. (2014) for a closed-cell stratocumulus case.

## 4 Budget analysis for evolution of LWP$_{\mathrm{c}}$

We perform a detailed budget analysis to understand the simulated LWP$_{\mathrm{c}}$ evolution. Previous studies used mixed-layer theory (MLT) to calculate the LWP$_{\mathrm{c}}$ tendency from the tendencies of BL mean liquid water potential temperature $(\theta_{\mathrm{l}})$ and total water mixing ratio $(q_{\mathrm{t}})$ as well as the motion of $z_{\mathrm{i}}$ (Wood, 2007; Caldwell and Bretherton, 2009; van der Dussen et al., 2014; Ghonima et al., 2015; Hoffmann et al., 2020). It is well-known that MLT is not applicable to the decoupled BL, which is prevalent in our simulations during the daytime. Here, we apply the MLT-based approach to both the BL and the "cloud volume" (CV), which we define for a given time $t$ as the volume consisting of all cloudy columns between $z_{\mathrm{i}}(t)$ and the first grid box interface below $\overline{z_{\mathrm{cb}}}$ (Figure S2). The choice of this volume is inspired by previous work showing success in assuming the cloud layer being well-mixed under decoupled conditions (Turton and Nicholls, 1987; Bretherton and Wyant, 1997). It is also based on our observation that in our simulations the entrainment velocity, diagnosed as

$$w_{\mathrm{e}} = \frac{\mathrm{d}z_{\mathrm{i}}}{\mathrm{d}t} - w_{\mathrm{s}}(z_{\mathrm{i}}), \tag{4}$$

is rarely negative, even at its weakest point in the late afternoon, meaning there is always some turbulent motion near the cloud top that mixes the air between the cloud layer and the FT. Different from previous work, we further focus on the cloudy region of the cloud layer. The specific definition of the CV base takes full advantage of quantities reported by SAM at the grid box interface to reduce the impacts of vertical interpolation. The CV depth defined this way is within a few percent of the actual cloud depth. We first show the derivation of CV budgets and then show results from both the BL and CV budgets.

### 4.1 Derivation

Consider a scalar quantity $\phi$ (in our case $\theta_{\mathrm{l}}$ or $q_{\mathrm{t}}$) at time $t$ in a volume consisting of a set of model columns covering a fraction of the domain area $(f)$ between the volume base height $z_0(t)$ and $z_{\mathrm{i}}(t)$. We denote the total amount of this scalar quantity and air mass in this volume with $\Phi$ and $M$, respectively. Since SAM solves the anelastic equations of motion, where the air density $\rho_0$ only changes with height,

$$\Phi = f(t) \int_{z_0(t)}^{z_{\mathrm{i}}(t)} \rho_0(z)\phi(z,t)\mathrm{d}z, \tag{5}$$



and

$$M = f(t) \int_{z_0(t)}^{z_i(t)} \rho_0(z) \mathrm{d}z = \langle \rho_0 \rangle f(t) h(t), \tag{6}$$

where $\langle \rho_0 \rangle$ is the mean air density of the volume, $\phi(z,t)$ is the time-dependent mean $\phi$ profile, and $h(t) = z_i(t) - z_0(t)$ is the volume thickness. The mean scalar quantity in this volume is

$$\langle \phi \rangle = \Phi/M. \tag{7}$$

Inspired by the derivation in Appendix B in Kazil et al. (2016), we build a budget for $\langle \phi \rangle$ from the budgets for $\Phi$ and $M$ via

$$\frac{\mathrm{d}\langle \phi \rangle}{\mathrm{d}t} = \frac{1}{M}\frac{\mathrm{d}\Phi}{\mathrm{d}t} - \frac{\langle \phi \rangle}{M}\frac{\mathrm{d}M}{\mathrm{d}t}. \tag{8}$$

The $\langle \phi \rangle$ tendency can also be decomposed into the contributions from various processes

$$\frac{\mathrm{d}\langle \phi \rangle}{\mathrm{d}t} = \sum_P \frac{\mathrm{d}\langle \phi \rangle}{\mathrm{d}t}\bigg|_P = \sum_P \left( \frac{1}{M}\frac{\mathrm{d}\Phi}{\mathrm{d}t}\bigg|_P - \frac{\langle \phi \rangle}{M}\frac{\mathrm{d}M}{\mathrm{d}t}\bigg|_P \right), \tag{9}$$

where the processes $P$ include volume-top entrainment (ENTR), processes at volume sides (LAT for lateral), radiation (RAD),
subsidence (SUBS), and processes at the volume base: transport flux at volume base (BASE), precipitation flux at volume base (PRCP), and a term tracking the impacts of the rising or lowering of the volume base (BM, standing for "base motion"). The $\mathrm{d}\langle \phi \rangle/\mathrm{d}t$ due to each of these seven processes can be calculated from $\mathrm{d}\Phi/\mathrm{d}t$ and $\mathrm{d}M/\mathrm{d}t$ due to the same process via Eq. (9).

When we apply this approach to the budget of $\langle \phi \rangle$ in a CV, $f$ is equivalent to cloud fraction $f_c$ and several terms are quite straightforward to estimate accurately. The RAD and BASE terms for $\Phi$ are directly computed from the 3-D modeled fields of
radiative heating rate, vertical velocity, and $\phi$, and neither process modifies $M$. Although we are dealing with non-precipitating cases, we retain the PRCP terms to minimize the residual. The BM term is calculated following

$$\frac{\mathrm{d}\langle \phi \rangle}{\mathrm{d}t}\bigg|_{\mathrm{BM}} = \frac{1}{M}\frac{\mathrm{d}\Phi}{\mathrm{d}t}\bigg|_{\mathrm{BM}} - \frac{\langle \phi \rangle}{M}\frac{\mathrm{d}M}{\mathrm{d}t}\bigg|_{\mathrm{BM}} = -\frac{\rho_0(z_0)\phi(z_0,t)f_c}{M}\frac{\mathrm{d}z_0}{\mathrm{d}t} + \frac{\rho_0(z_0)\langle \phi \rangle f_c}{M}\frac{\mathrm{d}z_0}{\mathrm{d}t}. \tag{10}$$

The SUBS term for $\Phi$ is diagnosed by applying the Reynolds Transport Theorem (RTT),

$$\begin{aligned}
\frac{\mathrm{d}\Phi}{\mathrm{d}t}\bigg|_{\mathrm{SUBS}} &= f_c \int_{z_0(t)}^{z_i(t)} \rho_0(z)\frac{\mathrm{d}\phi(z,t)}{\mathrm{d}t}\bigg|_{\mathrm{SUBS}} \mathrm{d}z + \rho_0(z_i)\phi(z_i,t)f_c\frac{\mathrm{d}z_i}{\mathrm{d}t}\bigg|_{\mathrm{SUBS}} \\
\qquad &= f_c \int_{z_0(t)}^{z_i(t)} \rho_0(z)\frac{\mathrm{d}\phi(z,t)}{\mathrm{d}t}\bigg|_{\mathrm{SUBS}} \mathrm{d}z + \rho_0(z_i)\phi(z_i,t)f_c w_s(z_i),
\end{aligned} \tag{11}$$

where $\mathrm{d}\phi(z,t)/\mathrm{d}t|_{\mathrm{SUBS}}$ is calculated by applying SAM's subsidence subroutine to the $\phi(z,t)$ profile. Note that although the CV base is defined to be close to $\overline{z_{\mathrm{cb}}}$, which evolves due to many processes, this choice of CV base is to avoid applying MLT later to deeper stratified layers. In other words, as long as the CV base sits in a well-mixed layer, there is no need to update





its height based on the cloud base height, and our choice to move it following the cloud base height is arbitrary. So, physical
processes do not directly move the CV base and there is no $dz_0(t)/dt$ in the terms for any processes but the BM term. The
SUBS term for $M$ is

$$\left.\frac{dM}{dt}\right|_{\mathrm{SUBS}} = \rho_0(z_\mathrm{i})f_\mathrm{c}w_\mathrm{s}(z_\mathrm{i}). \tag{12}$$

The ENTR flux of $\Phi$ can be parameterized as

$$\left.\frac{d\Phi}{dt}\right|_{\mathrm{ENTR}} = \rho_{0,\mathrm{e}}\phi_\mathrm{e}(t)f_\mathrm{c}w_\mathrm{e}, \tag{13}$$

where $w_\mathrm{e}$ is the entrainment velocity estimated from Eq. (4) and $\rho_{0,\mathrm{e}}$ and $\phi_\mathrm{e}(t)$ are an air density and a $\phi$ value that are relevant
to the entrainment flux of $\phi$. (Subscript "e" stands for "entrainment", as in $w_\mathrm{e}$.) Combined with the ENTR term for $M$, the
contribution of entrainment to the $\langle\phi\rangle$ tendency is

$$\left.\frac{d\langle\phi\rangle}{dt}\right|_{\mathrm{ENTR}} = \frac{1}{M}\left.\frac{d\Phi}{dt}\right|_{\mathrm{ENTR}} - \frac{\langle\phi\rangle}{M}\left.\frac{dM}{dt}\right|_{\mathrm{ENTR}} = \frac{\rho_{0,\mathrm{e}}\phi_\mathrm{e}(t)f_\mathrm{c}w_\mathrm{e}}{M} - \frac{\rho_0(z_\mathrm{i})\langle\phi\rangle f_\mathrm{c}w_\mathrm{e}}{M}. \tag{14}$$

Assuming constant $\rho_0$ and overcast conditions ($f_\mathrm{c} = 1$), Eq. (14) reduces to

$$\left.\frac{d\langle\phi\rangle}{dt}\right|_{\mathrm{ENTR}} = \frac{1}{h}w_\mathrm{e}\Delta\phi, \tag{15}$$

where $\Delta\phi$ is the $\phi$ jump at the volume top. Previous work used $\phi$ values at certain levels above and below $z_\mathrm{i}$ (usually denoted
as $z_+$ and $z_-$) to calculate the jump (Yamaguchi et al., 2011; Bretherton et al., 2013). Comparing Eqs. (14) and (15), it seems
that we can follow a similar method to find a level above $z_\mathrm{i}$ and use the $\phi$ and $\rho_0$ at this level in place of $\phi_\mathrm{e}$ and $\rho_{0,\mathrm{e}}$. However,
it is unclear what formula can be used to reliably find this level for all coupled and decoupled conditions in our simulations.
With Eq. (14), the challenging part is the entrainment flux term, $d\Phi/dt|_{\mathrm{ENTR}}$. For now, we approximate it with the entrainment
flux term for the BL. We first apply Eq. (9) to the whole BL. In this case, the BM and LAT terms vanish and the BASE term is
calculated from the surface fluxes reported by SAM (denoted with SURF term). Because all terms other than the ENTR term
are relatively easy to estimate directly and accurately, we don't keep a residual term, essentially lumping any residual into the
ENTR term. So,

$$\left.\frac{d\langle\phi\rangle_{\mathrm{BL}}}{dt}\right|_{\mathrm{ENTR}} = \frac{d\langle\phi\rangle_{\mathrm{BL}}}{dt} - \left(\left.\frac{d\langle\phi\rangle_{\mathrm{BL}}}{dt}\right|_{\mathrm{RAD}} + \left.\frac{d\langle\phi\rangle_{\mathrm{BL}}}{dt}\right|_{\mathrm{SUBS}} + \left.\frac{d\langle\phi\rangle_{\mathrm{BL}}}{dt}\right|_{\mathrm{SURF}} + \left.\frac{d\langle\phi\rangle_{\mathrm{BL}}}{dt}\right|_{\mathrm{PRCP}}\right). \tag{16}$$

Then,

$$\left.\frac{d\Phi_{\mathrm{BL}}}{dt}\right|_{\mathrm{ENTR}} = \langle\phi\rangle_{\mathrm{BL}}\left.\frac{dM_{\mathrm{BL}}}{dt}\right|_{\mathrm{ENTR}} + M_{\mathrm{BL}}\left.\frac{d\langle\phi\rangle_{\mathrm{BL}}}{dt}\right|_{\mathrm{ENTR}} = \rho_0(z_\mathrm{i})\langle\phi\rangle_{\mathrm{BL}}w_\mathrm{e} + M_{\mathrm{BL}}\left.\frac{d\langle\phi\rangle_{\mathrm{BL}}}{dt}\right|_{\mathrm{ENTR}}. \tag{17}$$

We use this term in place of $d\Phi/dt|_{\mathrm{ENTR}}$ in the CV budget.

Regarding the LAT term, we can write

$$\left.\frac{\langle\phi\rangle}{M}\frac{dM}{dt}\right|_{\mathrm{LAT}} = \frac{\langle\phi\rangle h(t)\langle\rho_0\rangle}{M}\frac{df_\mathrm{c}}{dt} = \frac{\langle\phi\rangle}{f}\frac{df_\mathrm{c}}{dt}. \tag{18}$$





Finally, we attribute all the remaining $\langle\phi\rangle$ tendency to $\mathrm{d}\Phi/\mathrm{d}t|_{\mathrm{LAT}}$ to close the budget without the need for a residual term.

Thus far, we have been tracking the budget of $\langle\theta_l\rangle$ and $\langle q_t\rangle$ and have not invoked MLT. Next, we apply the following equation for the LWP$_c$ tendency, derived based on MLT, to the CV,

$$\frac{\mathrm{d}\mathrm{LWP}_c}{\mathrm{d}t} = \Gamma_1 \langle\rho_0\rangle (z_i - z_{cb}) \left[ \frac{\mathrm{d}z_i}{\mathrm{d}t} - \left( \frac{\mathrm{d}z_{cb}}{\mathrm{d}\langle q_t\rangle} \frac{\mathrm{d}\langle q_t\rangle}{\mathrm{d}t} + \frac{\mathrm{d}z_{cb}}{\mathrm{d}\langle\theta_l\rangle} \frac{\mathrm{d}\langle\theta_l\rangle}{\mathrm{d}t} \right) \right], \tag{19}$$

where $z_{cb}$ is the mean cloud base height, $\Gamma_1$ is the liquid water adiabatic lapse rate, and $\mathrm{d}z_{cb}/\mathrm{d}\langle\theta_l\rangle$ and $\mathrm{d}z_{cb}/\mathrm{d}\langle q_t\rangle$ are based on the derivation in Ghonima et al. (2015) and follow similar notations in Hoffmann et al. (2020). In the calculation of $\Gamma_1$, $\mathrm{d}z_{cb}/\mathrm{d}\langle\theta_l\rangle$, and $\mathrm{d}z_{cb}/\mathrm{d}\langle q_t\rangle$, the actual cloud base air temperature and pressure are used. We decompose $\mathrm{d}z_i/\mathrm{d}t$ into the sum of $w_e$ and $w_s$, substitute $\mathrm{d}\langle q_t\rangle/\mathrm{d}t$ and $\mathrm{d}\langle\theta_l\rangle/\mathrm{d}t$ with the sum of individual budget terms diagnosed earlier, and finally group the $\mathrm{d}z_i/\mathrm{d}t$, $\mathrm{d}\langle q_t\rangle/\mathrm{d}t$, and $\mathrm{d}\langle\theta_l\rangle/\mathrm{d}t$ terms on the right-hand side of Eq. (19) by processes. Budget terms are diagnosed at the end of
each simulation hour (local time 40 min past each hour)

## 4.2  Diurnal cycles of BL budgets

We briefly introduce the diurnal cycles of the BL $\langle\theta_l\rangle$ and $\langle q_t\rangle$ budgets and the LWP$_c$ budget when they are used in Eq. (19) to provide a reference for the CV budgets in the next subsection.

The BL $\langle\theta_l\rangle$ and $\langle q_t\rangle$ budgets share similarity between the three categories, i.e., lo$\mathcal{D}$loL, hi$\mathcal{D}$loL, and hi$\mathcal{D}$hiL (Figure 5).
For the BL $\langle\theta_l\rangle$ budget (left column in Figure 5), RAD and ENTR are the leading terms during the nighttime. After sunrise, RAD quickly changes from cooling to warming, while ENTR warming weakens at a slower rate, leading to a peak in positive net BL $\langle\theta_l\rangle$ tendency in the morning. For the BL $\langle q_t\rangle$ budget (right column Figure 5), ENTR and SURF are the leading terms throughout the day. After sunrise, ENTR drying weakens faster than SURF moistening, leading to a peak in positive net BL $\langle q_t\rangle$ tendency between noon and 15:00 LT. Recall that in MLT, the subsidence has zero contributions to the tendencies of both
the mixed-layer $\langle\theta_l\rangle$ and mean $\langle q_t\rangle$. In our case, the contributions are not zero but still small compared with leading terms.

Figure 6 shows the LWP$_c$ budgets when the BL $\langle\theta_l\rangle$ and $\langle q_t\rangle$ budgets are used in Eq. (19). Comparing the actual LWP$_c$ tendency and the residual in the right column of Figure 6, applying MLT to the BL achieves fairly good closure during the nighttime for the lo$\mathcal{D}$loL category and between 02:00 and sunrise for the hi$\mathcal{D}$loL category; the residual continues to grow between 23:00 and sunrise for the hi$\mathcal{D}$hiL category. During the daytime, the residual is unacceptably large, demonstrating that
applying the MLT-based LWP$_c$ budget analysis to the BL is no longer appropriate.

The left column in Figure 6 shows the actual LWP$_c$ tendency as well as the contributions from the RAD, ENTR, SUBS, and SURF terms. During the nighttime, the most distinct feature is that the SUBS term is much more important relative to other terms in the LWP$_c$ budget than in the BL $\langle\theta_l\rangle$ and $\langle q_t\rangle$ budgets. This is due to the strong negative contribution by the subsidence to the $\mathrm{d}z_i/\mathrm{d}t$ term in Eq. (19). It is more negative for the hi$\mathcal{D}$loL and hi$\mathcal{D}$hiL categories because cases in these two categories
have a higher $z_i$ and thus a stronger subsidence due to the subsidence profile we impose. The ENTR term is comparable to other terms because its strong warming and drying effect (Figure 5) is offset by its positive contribution to the $\mathrm{d}z_i/\mathrm{d}t$ term. We do not discuss the results for the daytime due to the large residual.





### 4.3 Diurnal cycles of CV budgets

We first present the diurnal cycles of CV $\langle\theta_l\rangle$ and $\langle q_t\rangle$ budgets averaged by category (Figure 7). Similar to the BL budgets,
the ENTR and RAD terms are the leading terms for the CV $\langle\theta_l\rangle$ budget during the nighttime. Both weaken after sunrise, with
RAD cooling weakening faster. The ENTR warming decreases steadily towards late afternoon and becomes stronger before
sunset. The main difference from the BL budgets in the left column of Figure 5 is that RAD is mostly cooling during the
daytime because much of the warming effect by RAD occurs in the subcloud layer and is excluded in the CV $\langle\theta_l\rangle$ budget.
This warming strengthens the stratification of the subcloud layer, weakens the turbulent motion, and limits its impacts on the
CV. The remaining effects of this subcloud warming on the CV are accounted as transport in BASE and LAT terms. The RAD
cooling becomes stronger after around 09:00 or 10:00. It continues to strengthen through the rest of the day for the lo$\mathcal{D}$loL and
hi$\mathcal{D}$hiL categories, even though the LWP$_c$ does not recover until afternoon (Figure 3a). This trend is dominated by the trend in
CV-integrated radiative heating rates (not shown). For the hi$\mathcal{D}$loL category, there is a second weakening-strenghening cycle.
This is a signature of the rapid lowering of $\overline{z_{cb}}$ in this category as the stratiform parts of the clouds shrink and cumulus parts
dominate (see Section 3 and Figure 2e) and, as a result, the total radiative divergence for the CV is distributed over a deeper
layer. Note that due to subsidence and the growing of $z_i$, the FT in all our simulations becomes drier over time. (FT $q_t$ values at
the end of the simulations are between 64% and 85% of those at sunrise.) This effect likely also modulates the balance between
longwave cooling and shortwave absorption.

As the ENTR term for the CV $\langle\theta_l\rangle$ continues to decrease after the radiation passes its morning weakest point, the BASE-n-
LAT term starts to play a more significant role (left column in Figure 7). This term is defined as the sum of the BASE and LAT
terms. It represents the processes associated with the interface between the CV and the rest of the BL (i.e., CV base and lateral
sides). It shows an opposite trend from the RAD term and becomes the main term balancing the radiation in the afternoon.
This can be interpreted as follows: while there is not enough kinetic energy for mixing across the inversion base, the radiative
cooling in the CV still couples with the dynamics inside the BL.

For $\langle q_t\rangle$, the ENTR and BASE-n-LAT terms are the leading terms (right column of Figure 7). Unlike the BASE-n-LAT term
for the $\langle\theta_l\rangle$ budget, which can warm or cool the CV at different times, the BASE-n-LAT term mostly moistens the CV.

As mentioned before, the base motion (BM) term comes from the arbitrary choice of CV base height, although it is related
to the actual cloud base height evolution. When the BL is stratified, a rising CV base means the air mass near cloud base, which
has lower $\theta_l$ than the CV mean, is excluded from the CV. This results in an increase in $\langle\theta_l\rangle$ in the CV. Similarly we can infer
the sign of this term for $\langle\theta_l\rangle$ and $\langle q_t\rangle$ budgets under other conditions. This BM term is near zero during the nighttime when the
BL is close to being well-mixed. Its relative importance peaks between 13:00 and 15:00 for both $\langle\theta_l\rangle$ and $\langle q_t\rangle$ when the cloud
base averaged for all cases starts to lower, accompanying the recovery of LWP$_c$. The magnitudes of cooling and moistening
during this time are greater than the magnitudes of warming and drying between 09:00 and noon, primarily because the layer
near the cloud base is more stratified in the afternoon.

The SUBS term always warms and dries the CV. Its effect peaks in the early afternoon around the time when the clouds are
the thinnest.




## 4.4 Diurnal cycles of the LWP$_c$ budget

Figure 8 shows the LWP$_c$ budget by category, with the actual LWP$_c$ tendency and ENTR, RAD, SUBS, and BASE-n-LAT terms in the left column and the BM and residual terms in the right column. The PRCP terms are negligible and omitted.

We start with the terms in the right column. For all three categories, it is encouraging that the residual in the LWP$_c$ budget is fairly small. The improvement over the results based on the BL budgets (Figure 6) is dramatic for all three categories between sunrise and early afternoon; it is also evident for the hi$\mathcal{D}$hiL category during the nighttime. Although the BM term is overall not important until early afternoon, quantifying it for CV $\langle\theta_l\rangle$ and $\langle q_t\rangle$ budgets makes the LAT term (and thus the BASE-n-LAT term) slightly more accurate. Interestingly, the sum of the residual and BM term is even closer to zero. Qualitatively, the correlation between the BM term and the residual is expected considering that more stratified conditions simultaneously lead to a larger BM term and less applicability of MLT.

Moving to the terms in the left column of Figure 8, we know based on the small sum of the residual and BM term that the ENTR, RAD, SUBS, and BASE-n-LAT terms collectively explain the actual evolution of the LWP$_c$ very well until early afternoon. In particular, we can infer from the small sum of the residual and BM term that the sum of these four terms captures the reduction of LWP$_c$, most rapid for the hi$\mathcal{D}$hiL category and least for the lo$\mathcal{D}$loL category, in the morning, as is evident in the time series of the actual LWP$_c$ tendency.

The ENTR, RAD, and BASE-n-LAT terms are expected to be the leading terms simply based on their roles in the CV $\langle\theta_l\rangle$ and $\langle q_t\rangle$ budgets. By contrast with the results in Figure 6, the SUBS terms are less important relative to the ENTR term. This is because the $dz_i/dt$ term in Eq. (19) is constant in the two versions of LWP$_c$ budget but the $d\langle\theta_l\rangle/dt$ and $d\langle q_t\rangle/dt$ terms are strongly affected by the depth over which the volume-integrated forcing is distributed.

The SUBS term has the smallest diurnal fluctuation among the four terms. As a result, one can infer that the net effect of the ENTR, RAD, and BASE-n-LAT terms would approximately follow the trend of the actual LWP$_c$ tendency for each category. Among these three terms, the ENTR and RAD terms always begin to weaken right after sunrise. The BASE-n-LAT term remains near its maximum strength until 09:00 for the lo$\mathcal{D}$loL category, but it starts to weaken right after sunrise for the other two categories. This delay is likely the signature of better coupling with the surface. Due to this delay, although the rate of ENTR weakening for the lo$\mathcal{D}$loL category is slower than for the hi$\mathcal{D}$loL category, the combined negative effect from ENTR and BASE-n-LAT terms (pink dash-dotted lines) diminishes faster between sunrise and 09:40 for lo$\mathcal{D}$loL. Since the change in the RAD term from sunrise to between 09:00 and 10:00 is about the same between these two categories, the delayed decrease in the BASE-n-LAT term explains the slower LWP$_c$ reduction for the lo$\mathcal{D}$loL category. The weakening of the BASE-n-LAT term balances that of the ENTR term closely for the hi$\mathcal{D}$hiL category and the net effect (the pick dash-dotted lines) only weakens very slowly. As a result, the line for the RAD term is nearly parallel to the line for the actual LWP$_c$ tendency. Interestingly, when the actual LWP$_c$ tendency becomes the most negative in the morning for the lo$\mathcal{D}$loL and hi$\mathcal{D}$loL categories, its value is very close to the SUBS term, meaning the ENTR, RAD, and BASE-n-LAT terms sum to about zero. It is unclear whether this is by accident but this is different for the hi$\mathcal{D}$hiL category, where the actual LWP$_c$ tendency can be much more negative than the SUBS term, driven by the dramatic change in the RAD term.



To summarize, applying the MLT to the CV achieves satisfactory closure for the LWP$_c$ budget from nighttime to early afternoon. In the morning, the coupling to the surface, evident in the BASE-n-LAT term, explains the relatively smaller loss of LWP$_c$ for the lo$\mathcal{D}$loL category. The strong reduction of the RAD cooling causes the rapid reduction of LWP$_c$ for the hi$\mathcal{D}$hiL category. In the next section, we will use the budget analysis to understand the evolution of individual LES ensemble members,
not just the mean evolution by category.

## 5 Nighttime and daytime evolution of LES ensemble members

With the categorization of cases and the budget analysis presented, we can now examine the nighttime and daytime evolution of simulations in detail.

### 5.1 Nighttime evolution of individual cases

Figure 9 highlights several aspects of the nighttime evolution. Overall, the nighttime evolution is characterized by the establishment of a positive correlation between LWP$_c$ and a characteristic FT $q_t$. (Since subsidence is the only process that modifies the FT $q_t$ profile in our simulations, the characteristic FT $q_t$ is determined as follows. For a given time, we track the air mass at 20 m above $z_i$ back in time using the subsidence profile, Eq. (1), to calculate its height at the beginning of the simulation, and represent the current FT $q_t$ with the initial $q_t$ at that height.) This can be seen by comparing the trajectories, colored by FT
$q_t$, during the first three hours after the start of the simulations (Figure 9a) and during the three hours before sunrise (Figure 9b). It is also evident in the time series of the correlation coefficient between LWP$_c$ and FT $q_t$ (Figure 9c). At the beginning of each simulation, LWP$_c$ is determined by three of the six prescribed parameters: BL $\theta_l$, BL $q_t$, and $h_{mix}$. As a result of the random sampling of the initial conditions, it is largely uncorrelated with the FT $q_t$ even after we exclude cases based on criteria described in Section 2. FT $q_t$ acts as a boundary condition for the simulated clouds. It affects LWP$_c$ by modulating entrain-
ment drying and the downward longwave radiation reaching the cloud top, two effects that compete with each other (Eastman and Wood, 2018). Based on the way we specify FT $q_t$ profiles, the FT humidity controlling the longwave radiation positively correlates with the FT humidity that is relevant to the entrainment. For example, a case with a dry FT in our ensemble would experience greater entrainment drying; at the same time, it experiences strong radiative cooling because the FT is more transparent to longwave radiation. Although this strong radiative cooling favors high LWP$_c$, it also drives the clouds to entrain more,
potentially reducing LWP$_c$. The positive correlation between LWP$_c$ and FT $q_t$ in our simulations suggests that the entrainment effect dominates.

Figures 9d–f show the LWP$_c$ velocity, defined as the ratio between LWP$_c$ change and mean LWP$_c$ over a period of time, for the three hours before sunrise in LWP$_c$–$z_i$, $N_d$–$z_i$, and $N_d$–LWP$_c$ planes, where the locations of dots are based on states at sunrise. Most cases with LWP$_c$ less than 60 g m$^{-2}$ at sunrise gain LWP$_c$ during the three hours before sunrise (Figures 9d
and 9f). This qualitatively agrees with Hoffmann et al. (2020) and Glassmeier et al. (2021). However, the sign of the LWP$_c$ velocity is mixed for cases with greater LWP$_c$, where only 56% cases are gaining LWP$_c$. Among these cases, there is a weak negative correlation between $z_i$ and LWP$_c$ velocity, i.e., shallower/deeper BLs tend to see increasing/decreasing LWP$_c$ When





projected onto the $N_d$–$LWP_c$ plane (Figure 9f), cases with low $LWP_c$ and low $N_d$ mostly gain $LWP_c$, while cases losing $LWP_c$ only occur under high $LWP_c$ and high $N_d$ conditions. To some extent, this is consistent with the findings in Hoffmann et al.

(2020) and Glassmeier et al. (2021).

However, due to some potentially realistic yet complicated correlations among $LWP_c$, $N_d$, $z_i$, and FT $q_t$, we cannot simply attribute the correlation between $LWP_c$ velocity and $N_d$ to $N_d$. First, there is a positive correlation between $LWP_c$ and $N_d$ because we focus on the non-precipitating conditions and high $LWP_c$ cases are only possible if $N_d$ is sufficiently high to suppress precipitation (Figure 9f). Second, due to the positive correlation between $LWP_c$ and $z_i$ (deeper $z_i$ supporting higher

$LWP_c$, Figure 9d), there is also a positive correlation between $z_i$ and $N_d$ (notice very few cases in the upper left corner of Figures 9e). Similarly, because of the positive correlation between $LWP_c$ and FT $q_t$ (Figures 9b and 9c), there is a positive correlation between FT $q_t$ and $N_d$ (not shown).

We examine the correlation between radiative cooling and $LWP_c$ to assess the impacts of the positive correlation between FT $q_t$ and $LWP_c$ on the $LWP_c$ tendency (Figure 10). Recall that to calculate the RAD term for the $LWP_c$ budgets, we first calculate

the CV-integrated radiative heating rate, then assume it evenly distributes in the CV to calculate the RAD term for the CV $\langle\theta_l\rangle$ budget, and then use Eq. (19) to calculate the RAD term for the $LWP_c$. The CV-integrated radiative heating rate strongly depends on FT $q_t$ while the cloud-top temperature (approximated using the lowest temperature in the mean temperature profile for the CV) explains a small portion of its variance (i.e., lower cloud-top temperature associates with less integrated radiative cooling, Figure 10a). The sensitivity of the CV-integrated radiative heating rate to FT $q_t$ increases for FT $q_t$ below 3 g kg$^{-1}$.

More than 90% of cases have $LWP_c$ greater than 40 g m$^{-2}$ at this time and the emissivitiy of these clouds should have saturated (Garrett et al., 2002; Petters et al., 2012). (Our integrated radiative heating rate with FT $q_t$ of 4.5 g kg$^{-1}$, the FT $q_t$ estimated from Figure 2 in Petters et al. (2012) is very close to the saturated cloud-integrated radiative heating for longwave radiation in their Figure 1.) However, the RAD contribution to the CV $\langle\theta_l\rangle$ budget strongly and positively correlates with $LWP_c$ (filled circles in Figure 10b) due to correlation between $LWP_c$ and $\langle q_t\rangle$ as well as the scaling by CV depth. Earlier, we showed that the

MLT-based budget works well for the lo$\mathcal{D}$loL and hi$\mathcal{D}$loL categories during the nighttime (Figure 6). One might argue that it is more appropriate to assume the CV-integrated radiative heating rate is distributed from the surface to $z_i$. This scaling reduces the slope but not the sign of the correlation between the scaled RAD term and $LWP_c$ (hollow circles in Figure 10b). It is only when we use the CV-integrated radiative cooling rate scaled with $z_i$ in Eq. (19) that we find a positive correlation between the scaled RAD term for $LWP_c$ tendency and $LWP_c$ (hollow circles in Figure 10c; compare with hollow circles in Figure 10b).

The ratio between the scaled RAD term for the $LWP_c$ tendency and for the CV $\langle\theta_l\rangle$ tendency depends on $\Gamma_l$, $\langle\rho_0\rangle$, cloud depth, and $dz_{cb}/d\langle\theta_l\rangle$. Both the positive correlations between the cloud depth and $LWP_c$, as discussed in Hoffmann et al. (2020), and between other prefactors and $LWP_c$ (not shown) contribute to this change in the sign of the correlation. For the $LWP_c$ velocity, the division by $LWP_c$ itself further modifies the correlation and the slope between a budget term and $LWP_c$ (Figure 10d). In summary, not only the FT $q_t$ but also the $z_i$, the coupling state, and other factors (e.g., the prefactors in Eq. 19)

shape the correlation between the radiative contribution to $LWP_c$ tendency or velocity and the $LWP_c$.

We show the behavior of other terms for the $LWP_c$ tendency in Figure 11a. The BASE-n-LAT term positively contributes to the $LWP_c$ tendency. It negatively correlates with $LWP_c$ for greater $LWP_c$, but positively correlates with it for lower $LWP_c$, prob-





ably because cases with lower $LWP_c$ at sunrise, mostly in the lo$\mathcal{D}$loL category, have weaker boundary layer circulation. The

ENTR term negatively contributes to the $LWP_c$ tendency. It positively correlates with $LWP_c$ for greater $LWP_c$, but negatively
correlates with it for lower $LWP_c$. Compared with the RAD and BASE-n-LAT terms, this correlation suggests that, to the first
order, the entrainment is determined by the driving force for the turbulence, e.g., the radiative cooling and the boundary layer
circulation. The SUBS term negatively contributes to the $LWP_c$ velocity and positively correlates with $LWP_c$. After scaling
by $z_i$, the BASE-n-LAT, ENTR, and SUBS terms show a much tigher positive, negative, and negative correlation with $LWP_c$
(Figure 11b).

## 5.2 Daytime evolution of individual cases

Figures 12a and 12b show the most distinct feature of the daytime evolution of the individual cases. More decoupled cases tend
to lose $LWP_c$ more rapidly between sunrise and 12:00. For cases with $z_i$ greater than about 0.9 km, the positive correlation
between $LWP_c$ and $z_i$ at sunrise (dots in Figure 9b) becomes negative by 12:00 (dots in Figure 12a). In the afternoon, the $LWP_c$
recovers for most cases and a positive correlation between $LWP_c$ and $z_i$ is restored by the end of the simulation.

To understand the factors controlling the evolution of $LWP_c$ in the $LWP_c$–$z_i$ plane, we investigate the behavior of four groups
of cases with different properties: (1) lo$\mathcal{D}$loL cases with $LWP_c$ at sunrise between 75 and 90 g m$^{-2}$ (2) hi$\mathcal{D}$loL cases with
$LWP_c$ at sunrise in the same range (hi$\mathcal{D}$loL Group 1), (3) hi$\mathcal{D}$loL cases with $LWP_c$ at sunrise between 150 and 180 g m$^{-2}$
(hi$\mathcal{D}$loL Group 2), and (4) hi$\mathcal{D}$hiL cases with $LWP_c$ at sunrise between 240 and 300 g m$^{-2}$. Comparing Figures 12c and 12d, all
four groups develop negative slopes between $LWP_c$ and $z_i$ between sunrise and 09:40, the least negative for the lo$\mathcal{D}$loL group
and the most negative for the hi$\mathcal{D}$hiL group. Figure 13a shows the $LWP_c$ tendencies and budget terms for each case in these
four groups. The mean $LWP_c$ tendency between sunrise and 09:40 differs between groups, by $z_i$, and by degree of coupling.
For example, the loss of the $LWP_c$ is faster/slower for groups with higher/lower $LWP_c$ at sunrise; within each group, cases
with greater $z_i$ tend to lose $LWP_c$ faster; the hi$\mathcal{D}$loL Group 1 loses $LWP_c$ faster than the lo$\mathcal{D}$loL group. Across different $z_i$,
the RAD term positively correlates with the actual $LWP_c$ tendency and shows similar spread (Figure 13b). The variation of the
RAD term between groups is consistent with both the nighttime behavior of the RAD term (i.e., more positive RAD term for
low $LWP_c$ and low FT $q_t$, e.g., cases with higher $z_i$ in the lo$\mathcal{D}$loL group and hi$\mathcal{D}$loL Group 1; also see Figures 9b and 10c) and
the anticipated greater absorption of shortwave radiation for cases with higher $LWP_c$ (e.g., the hi$\mathcal{D}$hiL group). Unfortunately,
we do not have separate longwave and shortwave radiative output to quantify the relative importance of longwave cooling and
shortwave warming at this point. The ENTR and BASE-n-LAT terms are larger in magnitude than the RAD term (Figures 13c
and 13d). The SUBS term shows negative $z_i$-dependence with small differences between groups (Figure 13e). The sum of the
BM term and the residual is very small, compared with other terms and the actual $LWP_c$ tendency (Figure 13f). Based on these
results, it is reasonable to take the sum of the SUBS, the BM, the PRCP, and the residual terms as a baseline and investigate
how much the RAD, the ENTR, and the BASE-n-LAT terms drive the actual $LWP_c$ tendency to deviate from this baseline.
Figures 13g and 13h shows the sum of the RAD, the ENTR, and the BASE-n-LAT terms as well as the sum of the ENTR
and the BASE-n-LAT terms. Combined with the RAD term in Figure 13b, we conclude that the differences in $LWP_c$ tendency





between groups with different LWP$_c$ at sunrise are more associated with the RAD term, and the other details derive from a subtle balance between the RAD, ENTR, and BASE-n-LAT terms.

## 6 Discussion

In this section, we discuss an uncertainty in our budget analysis method, and then address the role of the interactive surface
fluxes in the simulations.

### 6.1 Uncertainty in ENTR term for $\langle\theta_l\rangle$ and $\langle q_t\rangle$ budgets

As described earlier, we use the entrainment fluxes (i.e., $\mathrm{d}\Phi/\mathrm{d}t|_{\mathrm{ENTR}}$) from the BL $\langle\theta_l\rangle$ and $\langle q_t\rangle$ budgets to calculate the ENTR term for the CV. However, because the cloudy region of a domain is more turbulent than the clear-sky region, one would expect a higher entrainment flux in the cloudy region than the domain-mean for partially cloudy scenes. Underestimating the
magnitude of entrainment fluxes for the CV budget will cause a compensating error in the BASE-n-LAT term because the latter holds the residual between the actual CV $\langle\theta_l\rangle$ and $\langle q_t\rangle$ tendencies and the sum of the other terms.

    In this subsection, we resort to the jump-based method (Eq. (15)) to assess the potential bias in our ENTR term. We first repeat the budget analysis for all clear-sky columns between the same base and top as the CV (denoted with "nCV", meaning "not CV"), and then partition the total entrainment warming and drying in the CV and the nCV with the cloudy region jump
$\Delta\phi_{\mathrm{CV}}$ and clear-sky jump $\Delta\phi_{\mathrm{nCV}}$. This alternative estimate of the entrainment tendency for the CV is

$$\left.\frac{\mathrm{d}\langle\phi\rangle}{\mathrm{d}t}\right|_{\mathrm{ENTR,alt}} = \frac{f_c\left(\mathrm{d}\langle\phi\rangle/\mathrm{d}t|_{\mathrm{ENTR}}\right) + (1-f_c)\left(\mathrm{d}\langle\phi\rangle_{\mathrm{nCV}}/\mathrm{d}t|_{\mathrm{ENTR}}\right)}{f_c + (1-f_c)\Delta\phi_{\mathrm{nCV}}/\Delta\phi_{\mathrm{CV}}}, \tag{20}$$

where "alt" stands for "alternative" and, again, $\phi$ represents either $\theta_l$ or $q_t$. The question becomes how to define $z_+$ and $z_-$ separately for $\phi$ profiles averaged in the cloudy and clear-sky regions to calculate the jumps. We follow Yamaguchi et al. (2011), where the authors check the domain-wide liquid water static energy ($s_l$) variance profile and define $z_+$ and $z_-$ as the
levels with $s_l$ variance falling to 5% of the peak value. This method works reasonably well for DYCOMS-II RF02, the case simulated in Yamaguchi et al. (2011). (See Appendix C in that work.) We apply a constant absolute $s_l$ variance threshold of 0.235 K$^2$ (5% of 4.7 K$^2$, the peak $s_l$ variance in Yamaguchi et al., 2011) to search for $z_+$ and $z_-$ to qualitatively capture the idea that the jump is smaller when turbulence mixing is weaker (lower peak $s_l$ variance).

    We take a few extra steps to handle potential outliers. We exclude all time steps with $f_c < 0.01$ (1.9% of all time steps) and
keep the entrainment tendencies with $f_c > 0.99$ unchanged. Sometimes, the peak $s_l$ variance of a profile (usually the clear-sky ones) is below 0.235 K$^2$ and no $z_+$ or $z_-$ are identified. For this situation, we keep a data point if only $\Delta\phi_{\mathrm{CV}}$ can be calculated (about 6.4% of all time steps) and set its $\Delta\phi_{\mathrm{nCV}}$ to 0, which actually exaggerates the difference between the cloudy and clear-sky region. We exclude a data point if neither $\Delta\phi_{\mathrm{CV}}$ nor $\Delta\phi_{\mathrm{nCV}}$ can be calculated, which rarely occurs.

    For all three categories, we find no significant difference between the current and the alternative ENTR terms until the
afternoon (Figure 14). These results certainly depend on details of our method, e.g., the value of the $s_l$ variance threshold.





However, without a more solid foundation for an alternative choice of the threshold, sensitivity tests would not provide more reliable quantification of the bias.

One other method is to partition the entrainment flux using Eq. (13), such that

$$\left.\frac{\mathrm{d}\Phi}{\mathrm{d}t}\right|_{\mathrm{ENTR,alt}} = \frac{1}{f_\mathrm{c} + (1-f_\mathrm{c})(\rho_{0,\mathrm{e}}\phi_\mathrm{e})_{\mathrm{nCV}}/(\rho_{0,\mathrm{e}}\phi_\mathrm{e})_{\mathrm{CV}}} \left.\frac{\mathrm{d}\Phi}{\mathrm{d}t}\right|_{\mathrm{ENTR}}. \tag{21}$$

If we use $\rho_0\phi$ at $z_+$ identified earlier as an estimate of $\rho_{0,\mathrm{e}}\phi_\mathrm{e}$, the resulting ENTR terms are even closer to our current estimates.

These results do not necessarily mean that our current ENTR term is accurate. They simply suggest that, the two alternative methods we test to introduce contrast between cloudy region and clear-sky entrainment produce limited "correction" to current ENTR estimates. While these results provide some confidence in the robustness of current ENTR estimates, it seems to be inconsistent with the argument that the cloudy region is more turbulent and thus should entrain more. We argue that this

inconsistency is partially rooted in the assumption that the movement of $z_\mathrm{i}$ is the result of the entrainment and the subsidence (Eq. (4)). We find that the air is on average descending/ascending at speeds around a few mm s$^{-1}$ near the mean $z_\mathrm{i}$ in the cloudy/clear-sky region, which are indeed at very similar heights, despite the mean updraft/downdraft for the bulk of BL in the cloudy/clear-sky region (Figure 14c). This is probably the signature of a mesoscale (instead of large-scale, e.g., the prescribed subsidence, which is horizontally uniform in the domain) mean circulation in the FT, similar to the one shown in Zhou and

Bretherton (2019). (See their Figure 9.) In other words, the cloudy/clear-sky region is more/less turbulent, but there may be a mesoscale downdraft/updraft limiting/promoting the growth of $z_\mathrm{i}$. With Eq. (4), the effect of this mesoscale mean air motion is lumped into the entrainment. This finding suggests that our current ENTR term should be interpreted as a *collective effect* of processes (other than the prescribed subsidence) that move the $z_\mathrm{i}$.

## 6.2 Response of surface fluxes to entrainment

One of the motivations for this work is to consider the role of the surface fluxes in compensating entrainment warming and drying. In this subsection, we take an analytical approach to this problem.

Considering that the evolutions of both the SHF and LHF in our simulations are driven primarily by $(\theta_{\mathrm{SST}} - \theta_{\mathrm{air}})$ and $(q_{\mathrm{sat}}(\mathrm{SST}) - q_{\mathrm{v,air}})$ and thus $\theta_{\mathrm{air}}$ and $q_{\mathrm{v,air}}$, we take the time-derivative on both sides of the two formulas in Eq. (3),

$$\frac{\mathrm{dSHF}}{\mathrm{d}t} \approx -C_T U \frac{\mathrm{d}\theta_{\mathrm{air}}}{\mathrm{d}t}, \quad \frac{\mathrm{dLHF}}{\mathrm{d}t} \approx -C_q U \frac{\mathrm{d}q_{\mathrm{v,air}}}{\mathrm{d}t}. \tag{22}$$

With a well-mixed BL, $\mathrm{d}\theta_{\mathrm{air}}/\mathrm{d}t$ and $\mathrm{d}q_{\mathrm{v,air}}/\mathrm{d}t$ on the right-hand sides of Eq. (22) should be close to the BL $\langle\theta_\mathrm{l}\rangle$ and $\langle q_\mathrm{t}\rangle$ tendencies. This assumption is supported by our simulations. To be specific, for time steps with small relative decoupling index ($\mathcal{D} < 1$), the SHF and LHF tendencies directly calculated from the time series of SHF and LHF agree well with those diagnosed using Eq. (22) by replacing $\mathrm{d}\theta_{\mathrm{air}}/\mathrm{d}t$ and $\mathrm{d}q_{\mathrm{v,air}}/\mathrm{d}t$ in that equation with the actual $\mathrm{d}\langle\theta_\mathrm{l}\rangle/\mathrm{d}t$ and $\mathrm{d}\langle q_\mathrm{t}\rangle/\mathrm{d}t$ (Figure 15a and 15b). Under these conditions ($\mathcal{D} < 1$), the response of SHF and LHF to entrainment warming and drying can be directly calculated

as

$$\left.\frac{\mathrm{dSHF}}{\mathrm{d}t}\right|_{\mathrm{ENTR}} = -C_T U \left.\frac{\mathrm{d}\langle\theta_\mathrm{l}\rangle}{\mathrm{d}t}\right|_{\mathrm{ENTR}}, \quad \left.\frac{\mathrm{dLHF}}{\mathrm{d}t}\right|_{\mathrm{ENTR}} = -C_q U \left.\frac{\mathrm{d}\langle q_\mathrm{t}\rangle}{\mathrm{d}t}\right|_{\mathrm{ENTR}}. \tag{23}$$





The magnitude of these responses is much greater than the actual SHF and LHF tendencies because entrainment is only one of the leading terms affecting $\theta_{\mathrm{air}}$ and $q_{\mathrm{v,air}}$ (Figure 15a and 15b).

From Eqs. (3) and (22), one can derive the time scale for the SHF to respond to changes in $\theta_{\mathrm{air}}$ to be

$$\tau_T \approx c_p \langle \rho_0 \rangle z_{\mathrm{i}} / C_T U, \tag{24}$$

where $c_p$ is the specific heat capacity. This formula is essentially the same as the surface flux component of the time scale in Eq. 1 in Bretherton et al. (2010) and based on Schubert et al. (1979b). With the data from our LES ensemble, $\tau_T$ ranges from 18 to 42 h with both mean and median around 30 h based on time steps with $\mathcal{D} < 1$. The time scales derived for the LHF to respond to changes in $q_{\mathrm{v,air}}$ are similar to $\tau_T$. In a hypothetical scenario when a shift in the BL temperature and moisture tendencies is initialized and dominated by enhanced entrainment warming and drying due to, e.g., aerosol perturbation, surface fluxes will not fully balance the additional warming and drying promptly. Assuming the changes in surface fluxes follow an exponential decay characterized by the aforementioned time scales, it takes about 1/3 of the time scale to offset about 15% of the additional warming and drying. To offset a more substantial portion, e.g., those reported by Chun et al. (2023), requires longer times or changes in $C_T$, $C_q$, or $U$.

Under more decoupled conditions ($\mathcal{D} > 2$), the actual SHF and LHF tendencies are quite different from those diagnosed from BL $\langle \theta_{\mathrm{l}} \rangle$ and $\langle q_{\mathrm{t}} \rangle$ tendencies (Figure 15c and 15d). The medians of the actual SHF tendencies are negative for most of the daytime and change to positive after 15 h. Those based on BL $\langle \theta_{\mathrm{l}} \rangle$ tendencies could be 1.1 W m$^{-2}$ h$^{-1}$ and 2.0 W m$^{-2}$ h$^{-1}$ more negative than the actual tendencies for the hi$\mathcal{D}$loL and hi$\mathcal{D}$hiL categories, respectively. This is because the net warming of the BL, mainly from entrainment and radiation, occurs aloft and has little impact on $\theta_{\mathrm{air}}$, which only increases due to the relatively weak SHF (left column in Figure 5). The medians of the actual LHF tendencies are much more negative than those based on BL $\langle q_{\mathrm{t}} \rangle$ tendencies. Here, entrainment drying in the upper part of BL should not be able to affect $q_{\mathrm{v,air}}$ very much, due to decoupling. As a result, the LHF moistens the surface-based mixed-layer faster than if the BL is well-mixed, which leads to faster decrease in LHF.

To summarize, the SHF and LHF always respond to the net changes in $\theta_{\mathrm{air}}$ and $q_{\mathrm{v,air}}$, contributed by all processes. Under well-mixed conditions, the time scale governing this response is relatively long. As a result, although the response of SHF and LHF to entrainment warming and drying can be estimated, it takes time and changes in surface transfer (characterized by $C_T$, $C_q$, and $U$) to see a strong response. Under decoupled conditions, the SHF and LHF can more quickly bring $\theta_{\mathrm{air}}$ and $q_{\mathrm{v,air}}$ towards equilibrium with the surface, given the shallow surface-based mixed-layer and assuming other factors remain unchanged. Thus, both SHF and LHF would weaken over time. This picture is compatible with our results for the BL $\langle q_{\mathrm{t}} \rangle$ budget, which is dominated by the ENTR and SURF terms (the right column in Figure 5). During the nighttime, there is no negative correlation between the ENTR and SURF terms, especially for the lo$\mathcal{D}$loL category, where the cases are more coupled towards sunrise. During the daytime, there is a negative correlation between these two terms but this can not be interpreted as the SURF term directly responding to the ENTR term due to decoupling.



## 7  Summary

In this work, we explore the impacts of diurnal cycles, free-tropospheric (FT) humidity values, and interactive surface fluxes on the cloud system evolution of non-precipitating marine stratocumuli by analyzing 245 cases in an LES ensemble generated by perturbing initial conditions.

We separate the cases into three categories with distinct behavior based on their relative decoupling index ($\mathcal{D}$) at 09:40 and cloud liquid water paths ($\mathrm{LWP_c}$) at sunrise: a lo$\mathcal{D}$loL category ($\mathcal{D} \leq 1$), a hi$\mathcal{D}$loL category ($\mathcal{D} > 1$ and $\mathrm{LWP_c} \leq 180$ g m$^{-2}$,

the highest $\mathrm{LWP_c}$ for the lo$\mathcal{D}$loL category), and a hi$\mathcal{D}$hiL category ($\mathcal{D} > 1$ and $\mathrm{LWP_c} > 180$ g m$^{-2}$). Cases in the lo$\mathcal{D}$loL category are commonly associated with lower $z_i$. They start with the lowest $\mathrm{LWP_c}$ and cloud fraction ($f_c$) among the three categories and may not ever become overcast. However, on average, they also experience the least reduction in $\mathrm{LWP_c}$ and $f_c$ during the daytime. Clouds in the hi$\mathcal{D}$loL category occur in deeper BLs, start with more $\mathrm{LWP_c}$, and tend to be overcast during the nighttime. On average, they experience dramatic $\mathrm{LWP_c}$ and $f_c$ reductions during the day. These clouds tend to evolve into

a cumulus-rising-into-stratocumulus structure in the afternoon. Clouds in the hi$\mathcal{D}$hiL category share many features with those in the hi$\mathcal{D}$loL category but show different timing and amplitude of daytime $\mathrm{LWP_c}$ and $f_c$ fluctuations. The diurnal cycles of $\mathrm{LWP_c}$ and $f_c$ for three categories are closely related to the diurnal cycles of their coupling states.

We perform a budget analysis to understand the diurnal cycle of $\mathrm{LWP_c}$ by tracking the mean $\theta_l$ and $q_t$ budgets for the "cloud volume" (CV), a volume consisting of all cloudy columns between the first grid box base below the mean cloud base and $z_i$,

and then applying the LWP budget equation (Eq. (19)) to the CV, assuming it is well-mixed. By focusing on the cloudy region of the cloud layer, this method closes the budget with a very small residual until early afternoon. In particular, it adequately captures the rapid $\mathrm{LWP_c}$ reduction in the morning for all categories. A delayed decrease in the positive contribution to $\mathrm{LWP_c}$ from the BASE-n-LAT term, a term that tracks the impacts of the processes associated with the interface between the CV and the rest of the BL (i.e., CV base and lateral sides), after sunrise explains the slower $\mathrm{LWP_c}$ reduction in the lo$\mathcal{D}$loL category

than in the hi$\mathcal{D}$loL category. For the hi$\mathcal{D}$hiL category, the strong decrease in the radiative (RAD) cooling results in the most rapid $\mathrm{LWP_c}$ reduction in this category.

The impact of a humid FT on the evolution of simulations during the nighttime is distinct. A positive correlation between FT $q_t$ and $\mathrm{LWP_c}$ emerges and strengthens towards sunrise. Because the longwave emissitivity of clouds is saturated in most cases, the FT $q_t$ strongly affects the CV-integrated radiative heating rate. As a result, there is stronger radiative cooling for

cases with lower $\mathrm{LWP_c}$ through the correlation between the FT $q_t$ and $\mathrm{LWP_c}$. This illustrates how the covariability among state variables and cloud controlling factors modifies the distribution of $\mathrm{LWP_c}$ tendency in state variable spaces. During the daytime, clouds in deeper BLs lose $\mathrm{LWP_c}$ faster in the morning, again suggesting that state variables beyond $\mathrm{LWP_c}$ and $N_d$ are necessary to understand the $\mathrm{LWP_c}$ tendency. A closer analysis reveals that the $\mathrm{LWP_c}$ tendency in the morning varies with the $\mathrm{LWP_c}$ at sunrise, $z_i$, and the degree of decoupling. A budget analysis for $\mathrm{LWP_c}$ shows that the subsidence term (SUBS) causes

a more negative $\mathrm{LWP_c}$ tendency at deeper $z_i$ and this effect is similar for cases with different $\mathrm{LWP_c}$ at sunrise and degree of decoupling. The entrainment (ENTR) and BASE-n-LAT terms closely balance each other, and there is a weak dependence of the net effect on $z_i$. It is the RAD term that differentiates cases with similar $z_i$ in terms of the $\mathrm{LWP_c}$ tendency.





We show that the surface flux fluctuations in our simulations are dominated by the evolution of the lowest model level air temperature ($\theta_{\text{air}}$) and water vapor mixing ratio ($q_{\text{v,air}}$), not the surface wind speed used in the surface flux calculation ($U$) or the

transfer coefficients ($C_T$ and $C_q$). As a result, the surface flux response to entrainment depends on the entrainment's impacts on $\theta_{\text{air}}$ and $q_{\text{v,air}}$. Under well-mixed conditions, this time scale for this response to offset entrainment warming and drying is $\sim \mathcal{O}(30\,\text{h})$, consistent with the timescale reported in Schubert et al. (1979a). Based on this finding, we estimate that it takes about 10 h for the surface fluxes to offset 15% of the changes in entrainment warming and drying, assuming no changes in transfer coefficients ($C_T$ and $C_q$) or surface wind speed ($U$); the magnitude of this response can be calculated from MLT-based

budget analysis. Under decoupled conditions, the surface fluxes do not respond directly to entrainment (by definition), although there could be a negative correlation between the time series of surface fluxes and entrainment.

We demonstrate the emergence of the correlations among environmental conditions and state variables as the clouds evolve. All these correlations project onto the correlations with $N_{\text{d}}$ and need to be carefully considered when we distill the causality between $N_{\text{d}}$ and variables like the LWP$_{\text{c}}$ tendency or the LWP$_{\text{c}}$ velocity. We pursue this task in Zhang et al. (2024).

**Appendix A:  Constructing initial thermodynamic profiles**

In this appendix, we describe the method for (1) creating the upper air $\theta_{\text{l}}$ and $q_{\text{t}}$ profiles and (2) connecting them with the initial BL $\theta_{\text{l}}$ and $q_{\text{t}}$ profiles (described in Section 2) to construct the initial $\theta_{\text{l}}$ and $q_{\text{t}}$ profiles.

To prepare for the upper air profiles, we generate ERA5-based climatological profiles in a few steps. First, we produce mean profiles from all ERA5 profiles in the Californian stratocumulus region (i.e., the 10° by 10° box between 20°N, 30°N, 120°W,

and 130°W as defined in Klein and Hartmann, 1993) during April, May, and June (the months with highest stratocumulus cover in the region; Wood, 2012) from 2000 to 2011. Then, we search for the height with the maximum $\theta_{\text{l}}$ gradient below 2 km and keep the mean profile segments between this height and 35.8 km, the top of the mean profiles.

When we connect the $\theta_{\text{l}}$ climatological profile produced this way to the initial BL profiles, some simulations experience very rapid growth in the inversion base height ($z_{\text{i}}$) in the first few hours, suggesting that the $\theta_{\text{l}}$ gradient across the inversion

is too weak. To solve this issue, we prepare a transitional profile for $\theta_{\text{l}}$. We average the observed $\theta_{\text{l}}$ profiles during the warm season legs of the MAGIC campaign after translating them vertically to line up at inversion bases and having their BL values subtracted at all heights. We keep the first 1.5 km of this mean profile above the inversion base.

To construct an initial $\theta_{\text{l}}$ profile, we first translate the transitional profile so that its lowest point attaches to point right above the inversion base. Next, we scale the ERA5-based $\theta_{\text{l}}$ climatological profile so that its lowest point attaches to the highest point

of the transitional profile (now sitting at 1.5 km above $h_{\text{mix}}$) while its highest point stays fixed at 35.8 km. For an initial $q_{\text{t}}$ profile, we scale the ERA5-based $q_{\text{t}}$ climatological profile so that its lowest point directly attaches to the point right above the inversion base while its highest point stays fixed at 35.8 km. A constant surface pressure of 1018.52 mb, based on ERA5 climatology, is used for all initial profiles. See Figure A1 for an illustration.



*Code and data availability.* The System for Atmospheric Modeling (SAM) code is publicly available at http://rossby.msrc.sunysb.edu/SAM.
html. The ERA5 data is archived at Copernicus Climate Change Service (C3S) Climate Data Store (CDS) (Hersbach et al., 2017). The
MAGIC data is available via ARM Data Discovery (Atmospheric Radiation Measurement (ARM) user facility, 2012). Data for reproducing
the results will be provided following acceptance.

*Author contributions.* GF, TY, and YC initiated this study. TY, FG, and YC designed the LES ensemble. TY kindly performed the sim-
ulations. YC analyzed the data and wrote the manuscript. All authors contributed throughout the study and provided comments on the
manuscript.

*Competing interests.* At least one of the (co-)authors is a member of the editorial board of Atmospheric Chemistry and Physics. Other than
this, the authors declare that they have no conflict of interests.

*Acknowledgements.* This study has been supported by the U.S. Department of Energy (DOE), Office of Science, Office of Biological and
Environmental Research, Atmospheric System Research (ASR) program (Interagency Award Number 89243023SSC000114), the U.S. De-
partment of Commerce (DOC), National Oceanic and Atmospheric Administration (NOAA), Climate Program Office, Earth's Radiation
Budget (ERB) program (Award Number 03-01-07-001), and NOAA cooperative agreements (NA17OAR4320101 and NA22OAR4320151).
FG acknowledges support from The Branco Weiss Fellowship - Society in Science, administered by ETH Zürich. The computational and
storage resources are provided by the NOAA Research and Development High Performance Computing Program (https://rdhpcs.noaa.gov).
We thank Marat Khairoutdinov for graciously providing the SAM model, Ryuji Yoshida for compiling the ERA5 climatology, and Jan Kazil
for insights regarding the budget analysis.





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



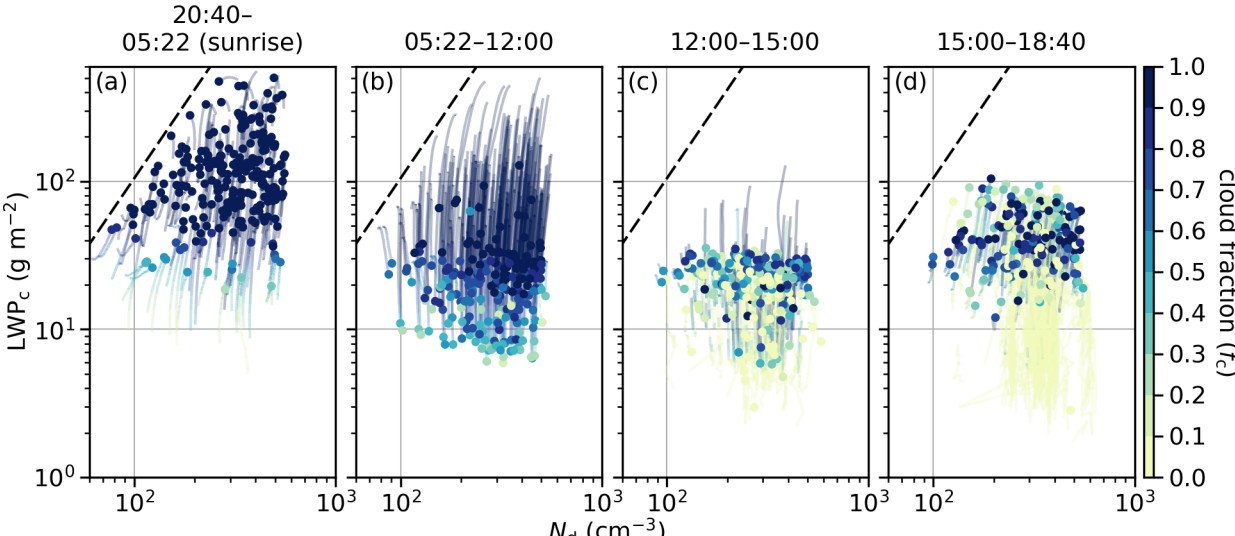

**Figure 1.** Evolution of the simulations in the plane of cloud droplet number concentration ($N_d$) and cloud LWP (LWP$_c$), split in to four time periods as shown in the panel titles. Curves indicate the trajectories over the time period and dots indicate the states at the end of the time period. The thick black dashed lines correspond to a characteristic mean drop radius of 12 $\mu$m, below which precipitation is inhibited.





**Figure 2.** (a) Relative decoupling index ($\mathcal{D}$) at 09:40 and (b) minimum cloud fraction (min $f_c$) after sunrise in the plane of inversion base height ($z_i$) and cloud LWP ($LWP_c$) at sunrise; (c) categories based on $\mathcal{D}$ at 09:40 and $LWP_c$ at sunrise: (1) lo$\mathcal{D}$loL ($\mathcal{D} \leq 1$), (2) hi$\mathcal{D}$loL ($\mathcal{D} > 1$ and $LWP_c \leq 180$ g m$^{-2}$), and (3) hi$\mathcal{D}$hiL ($\mathcal{D} > 1$ and $LWP_c > 180$ g m$^{-2}$); time series of (d) median and quantiles of $\mathcal{D}$ and (e) medians of $z_i$, $\overline{z_{cb}}$, and $\overline{z_{LCL}}$ by category. The vertical dashed grid lines in Panels (d) and (e) indicate sunrise.



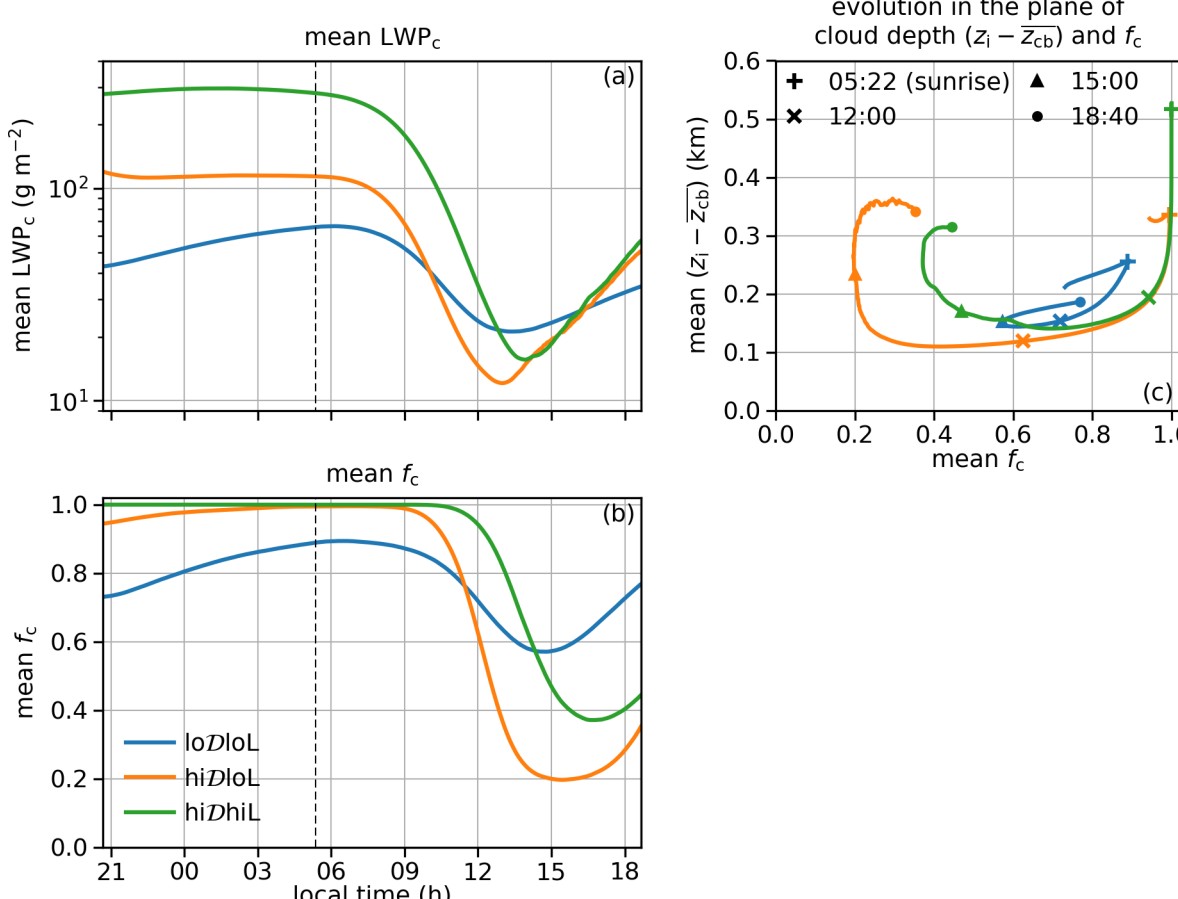

**Figure 3.** Time series of (a) mean cloud LWP ($LWP_c$), (b) mean cloud fraction ($f_c$); and (c) evolution by category in the plane of cloud depth $(z_i - \overline{z_{LCL}})$. The vertical dashed grid lines in Panels (a) and (b) indicate sunrise.





**Figure 4.** Time series of (a) surface sensible heat flux (SHF), (b) surface latent heat flux (LHF), (c) difference between potential temperature based on sea surface temperature ($\theta_{SST}$) and lowest model level air potential temperature ($\theta_{air}$), (d) difference between saturation mixing ratio at SST ($q_{sat}(SST)$) and lowest model level water vapor mixing ratio ($q_{v,air}$), (e) transfer coefficient for SHF ($C_T$), and (f) wind speed used for surface fluxes calculation ($U$) by category. The vertical dashed grid lines indicate sunrise.



**Figure 5.** Time series of actual BL $\langle \theta_l \rangle$ and $\langle q_t \rangle$ tendencies and budget terms due to individual processes by category. The vertical dashed grid lines indicate sunrise.

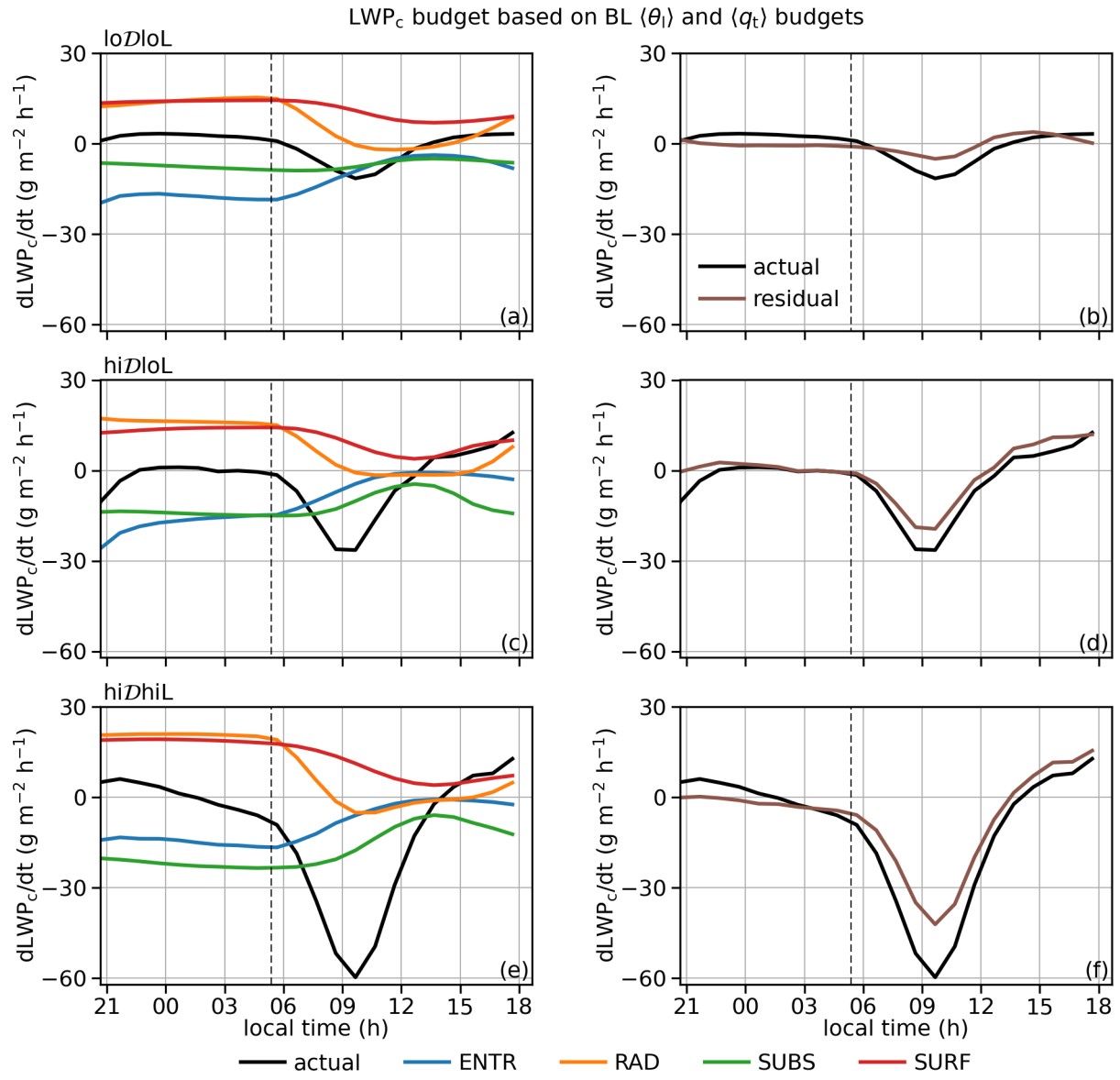

**Figure 6.** Time series of LWP$_c$ tendencies and budget terms due to individual processes by category, based on BL $\langle\theta_l\rangle$ and $\langle q_t\rangle$ budgets. The actual LWP$_c$ tendencies are shown in both the left and right columns for easier comparison with individual budget terms. The vertical dashed grid lines indicate sunrise.




**Figure 7.** Time series of actual CV $\langle \theta_l \rangle$ and $\langle q_t \rangle$ tendencies and budget terms due to individual processes by category. The vertical dashed grid lines indicate sunrise.

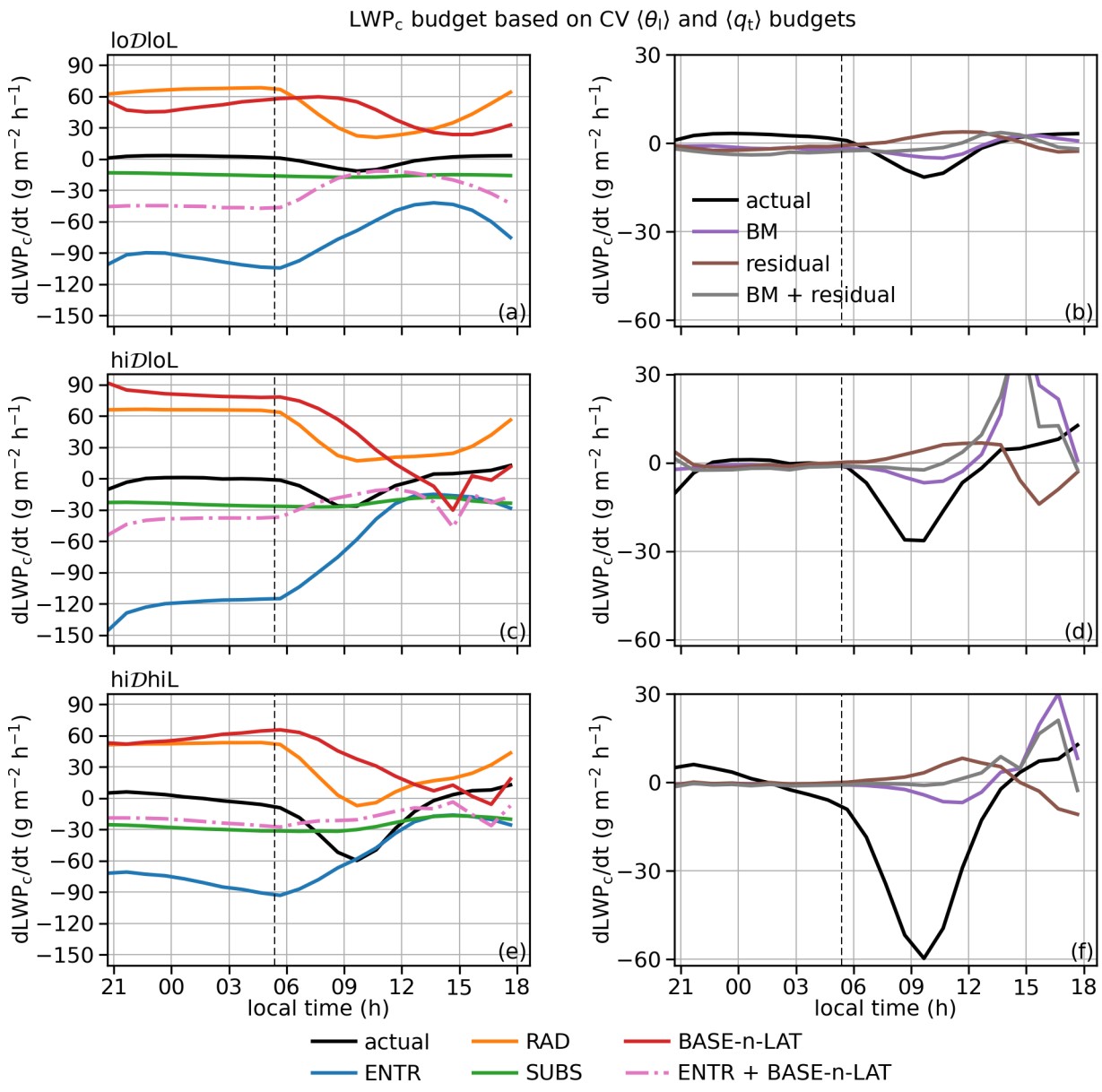

**Figure 8.** Time series of $LWP_c$ tendencies and budget terms due to individual processes by category, based on CV $\langle\theta_l\rangle$ and $\langle q_t\rangle$ budgets. The actual $LWP_c$ tendencies are shown in both the left and right columns for easier comparison with individual budget terms. The vertical dashed grid lines indicate sunrise.







**Figure 9.** Evolution of LES ensemble members during nighttime. In Panels (a) and (b), curves indicate the trajectories over the time period, and dots indicate the states at the end of the time period shown in the panel titles. "SR" indicates sunrise.





**Figure 10.** Radiative cooling at 04:40 LT. (a) CV-integraged radiative heating rate, (b) RAD term for CV $\langle\theta_l\rangle$ budget, (c) RAD term for $LWP_c$ budget, (d) radiative contribution to $LWP_c$ velocity. Hollowed circles in Panels (b) and (c) represent the tendencies when the CV-integraged radiative heating rate is hypothetically uniformly distributed over the entire BL depth.






**Figure 11.** A few extra terms for LWP$_c$ budget at 04:40 LT, in addtion to the RAD term in Figure 10c.



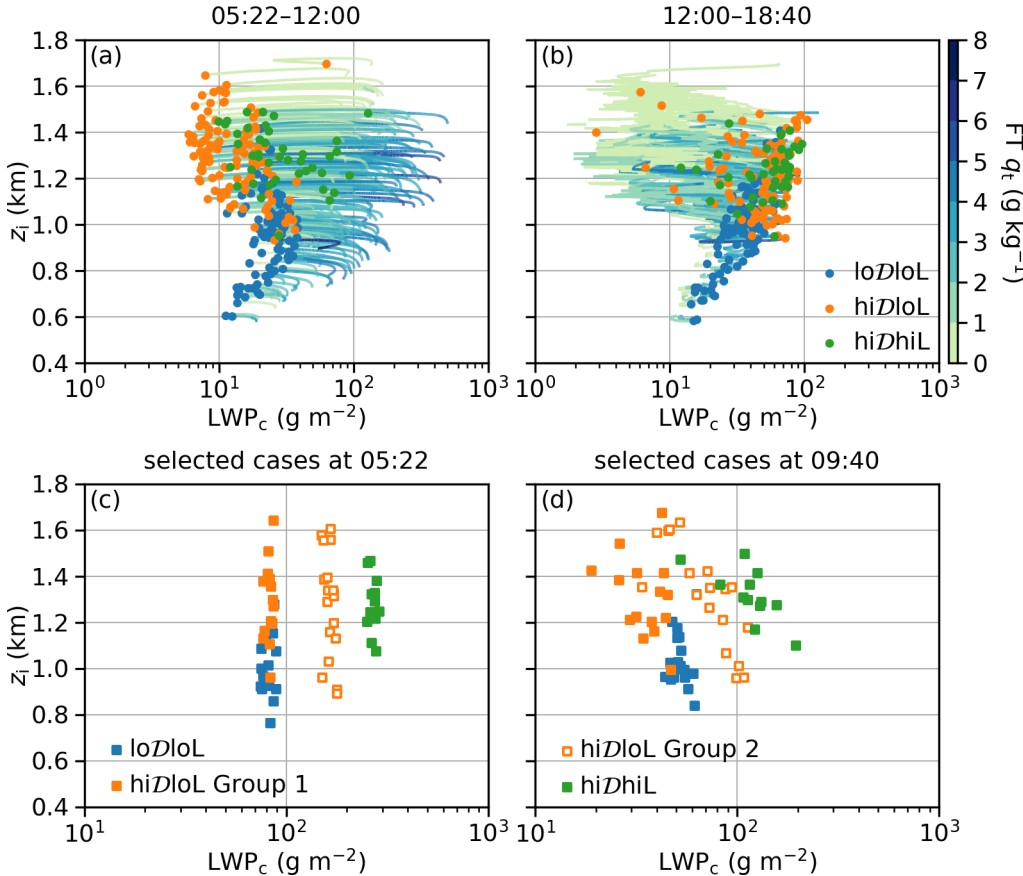

**Figure 12.** Evolution of LES ensemble members during daytime. In Panels (a) and (b), curves indicate the trajectories over the time period and dots indicate the states at the end of the time period, shown in the panel titles. Symbols in Panels (c) and (d) indicate groups of cases that are selected for further examination: (1) lo$\mathcal{D}$loL cases with LWP$_c$ at sunrise between 75 and 90 g m$^{-2}$ (2) hi$\mathcal{D}$loL cases with LWP$_c$ at sunrise in the same range (hi$\mathcal{D}$loL Group 1), (3) hi$\mathcal{D}$loL cases with LWP$_c$ at sunrise between 150 and 180 g m$^{-2}$ (hi$\mathcal{D}$loL Group 2), and (4) hi$\mathcal{D}$hiL cases with LWP$_c$ at sunrise between 240 and 300 g m$^{-2}$.





**Figure 13.** Mean LWP$_c$ tendencies and budget terms due to individual processes for selected cases between sunrise and 09:40.

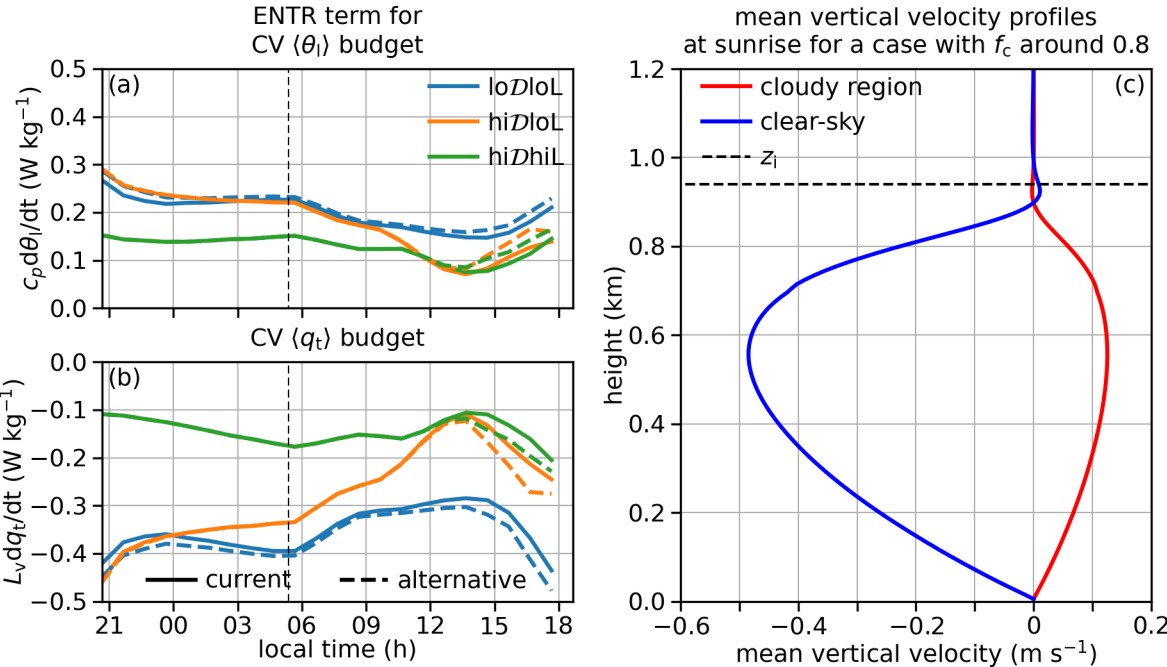

**Figure 14.** Time series of current and alternative estimates of the entrainment contribution to CV (a) $\langle\theta_l\rangle$ and (b) $\langle q_t\rangle$ budgets. The vertical dashed grid lines indicate sunrise. Panel (c) shows an example to facilitate the discussions near the end of Section 6.1.



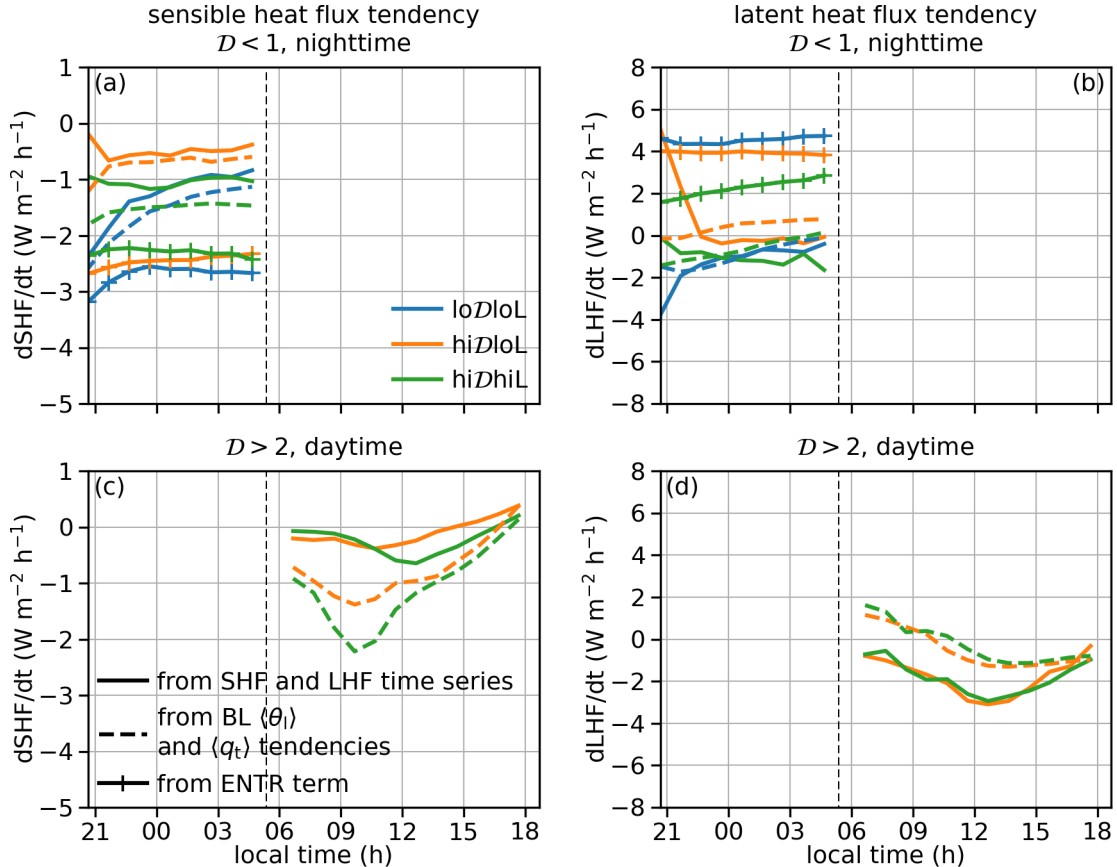

**Figure 15.** Time series of sensible heat flux (SHF, left column) and latent heat flux (LHF, right column) tendencies (1) directly from time series of SHF and LHF (solid lines), (2) calculated from actual BL $\langle\theta_l\rangle$ and $\langle q_t\rangle$ tendencies (dashed lines), and (3) contributed from the ENTR term for the BL $\langle\theta_l\rangle$ and $\langle q_t\rangle$ tendencies (solid lines with "+") under more coupled condition (relative decoupling index $(\mathcal{D}) \leq 1$, upper row) and more decoupled condition $(\mathcal{D} > 2$, lower row). The vertical dashed grid lines indicate sunrise.



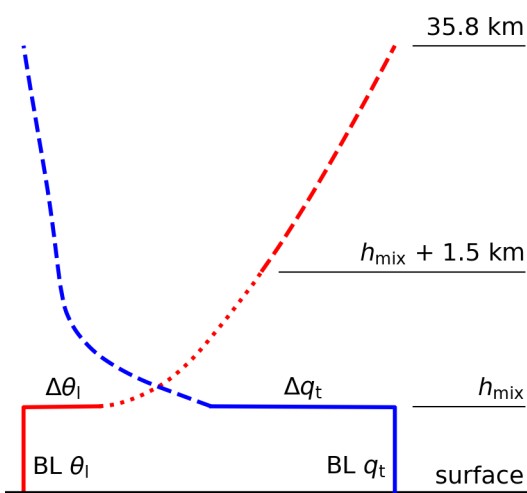

**Figure A1.** A sketch showing the construction of initial $\theta_l$ and $q_t$ profiles (in red and blue, respectively) from initial BL profiles (solid segments), ERA5-based climatological profiles (dashed segments) and the MAGIC-based transitional $\theta_l$ profile (dotted segment).