# Peer review of "Diurnal evolution of non-precipitating marine stratocumuli in an LES ensemble"

_EGUsphere, 2024_

## Author Comment (AC1)

We thank the reviewers for their constructive comments.

Before we respond to these comments, we would like to note that: To address some comments, we found it necessary to re-run the whole ensemble with an new SST configuration as a sensitivity test. There have been some updates in the programming environment on our server between the initial submission and the time when we re-ran the simulations. As a result, the compiler we currently have (Intel 2023) is different from the one used for the ensemble in the initial submission (Intel 2018). To maintain consistency in the programming environment for simulations with the original and the new SST configurations, we also re-ran the whole ensemble with the original SST configuration using the new compiler. The simulations with the original SST configuration produced by Intel 2018 and 2023 compilers are similar. In other words, all our original results qualitatively hold. Both of our revised manuscript and response are based on simulations produced with the new compiler.

In the rest of this file, we show the comments from Reviewers #1 and #2 in red and blue, respectively, and our response in black.

**Reviewer #1**

https://doi.org/10.5194/egusphere-2024-1033-RC1

Summary:

An ensemble of large eddy simulations of marine stratocumulus clouds is run and analyzed for a variety of idealized summertime conditions over the Northeast Pacific Ocean. Building on earlier work by led by co-authors Hoffman and Glassmeier, the new simulations use a more realistic radiation parameterization, including a diurnal cycle, as well as interactive surface fluxes. The analysis seeks to understand how the diurnal cycle of cloud cover and LWPc (in-cloud liquid water path) are determined by the base state (sorting into bins with high/low decoupling and LWPc) and by different processes in understanding how different processes fix LWPc. Highlights include: daytime reductions in cloud fraction and LWPc are larger in more decoupled boundary layers. Larger free tropospheric humidity tends to support higher LWPc during nighttime.

Assessment:

The paper is well-written and nicely tells a story about how stratocumulus-capped boundary layers vary across the diurnal cycle. The manuscript is interesting and compelling, but I was struck that the paper focuses much more strongly on LWPc than on cloud fraction, which is presumably a stronger control on shortwave cloud radiative effects during the diurnal cycle in many of these simulations. However, I expect that another paper based on these simulations will tell that story later. I think the manuscript is a good fit for ACP and should be published, though I would ask the authors to consider the suggestions I make below along with those of the other reviewers.

Recommendation: Minor Revisions

============================

Major comment:

1. Regarding the LWPc budget:

1a. For the domain-mean budget, it is natural for the subsidence term to be based on the prescribed, large-scale subsidence. However, for the cloud-volume (CV) budget, I find it puzzling that the mean vertical velocity in cloudy columns was not used. I did not ponder this until the end of section 6.1 (on the uncertainty in the entrainment term). If the mean velocity profile in the cloudy columns was not saved, then keep the analysis as it is, but if not, wouldn't the analysis be more robust if the mean cloudy-column subsidence was used for the SUBS_CV term? Then, the ENTR_CV term would be based on the actual entrainment rate in the cloudy-columns, which might be more closely tied to the radiative cooling there. The authors are free to argue that the current setup is more compelling and/or that the differences are only important in the afternoon when the cloud fraction falls precipitously in some of the simulations, but I wanted to suggest this.

By using the domain-mean, instead of CV-mean or cloudy-column mean, vertical velocity profile to represent the subsidence, we essentially define the large-scale subsidence as a mean vertical motion that is superimposed on the vertical velocity simulated by the LES. In other words, there is a conceptual separation between what is considered "large scale" (a mean vertical motion that is horizontally homogeneous across the domain) and what is "meso" and smaller scales (all other air

motions that are inhomogeneous within the domain). This is also how the subsidence is typically prescribed for LES.

Figure 14c in the original manuscript (Figure 13c in the revised manuscript) showed an example of cloudy-column mean and clear-sky mean vertical velocity profiles. We interpret features in these profiles as the updraft and downdraft branches of in-domain mesoscale circulations, not subsidence.

Then the question is what vertical motion contributes to entrainment. We argue that the air motion across many scales other than the subsidence may contribute to entrainment. Yamaguchi and Randall 2012 showed that the strongest downdrafts that entrain FT air deep into the BL often occur at cloud edge for a broken cloud field. Zhou and Bretherton 2019 also clearly showed a composite down-slope flow that moves from the thicker part of a stratocumulus to the thinner part, eventually penetrating into the BL. Given the above reasoning, we estimate entrainment velocity as the difference between $dz_i/dt$ and the domain-wide subsidence.

1b. The LWPc budget involves large terms with different signs (e.g., RAD, ENTR, BASE-n-LAT), with the important d(LWPc)/dt trend being the small residual after these large terms almost --- but not exactly --- offset each other. Since it's hard to visualize the result of such cancellation, I found myself seeking some way to understand the budget better. From this, I had two (very optional) suggestions:

- Instead of decomposing the dz_i/dt term into separate w_s and w_e as specified on line 272-273, include the whole d(z_i)/dt term in the budget. This is easier for the reader to understand and avoids another pair of opposing terms in the budget. Helpfully, this term switches sign from night to day in some of the simulations, so that changes in dz_i/dt --- driven here by changes in entrainment --- may play a role in the diurnal cycle of LWPc. Doing this would modify the ENTR term in the LWPc budget, since it would only represent the entrainment impacts on cloud base height through d<qt>/dt_CV and d<thetal>/dt_CV. Only make such a change if the authors find it clarifies the story about the budgets.

We agree that there could be different ways of grouping terms. For example, in Hoffmann et al. 2020JAS, a single "cloud-top motion" term was used to represent the impacts of $dz_i/dt$ on LWP. To address the reviewer's comments, we separate each of the ENTR and SUBS terms for the $LWP_c$ budget into two parts: warming/drying and $dz_i/dt$. Then we report a combined term associated with $dz_i/dt$. As the reviewer suggested, this latter term avoids another pair of opposing terms in the budget. But then we end up with a SUBS term only for its warming/drying effect. This adds more lines to the plot.

Randall 1984 showed that a stratocumulus may gain LWP from entrainment if its effect on $z_i$ outweighs the its warming and drying effect. We choose to group all effects of entrainment together as a total ENTR term to see the net effect.

We now include two figures in the supplementary material (Figures S5 and S7) to show the decomposed ENTR and SUBS terms as well as the combined $dz_i/dt$ term.

- (MORE SPECULATIVE) Re-group the terms, combining exchange within the boundary layer (BASE-n-LAT) with radiation (RAD) since both drive increases in LWPc. Both can also be seen as driving entrainment, both the familiar cloud top cooling through (RAD) and fluxes through cloud base

(BASE). Write ENTR = - C_ENTR * (RAD + BASE-n-LAT), where C_ENTR is something like the entrainment drying efficiency of the (RAD + BASE-n-LAT) terms. Then, the budget for LWPc is dominated by SUBS and (1-C_ENTR)*(RAD + BASE-n-LAT), with variations in C_ENTR controlling whether LWPc grows or decays. Looking at the budgets in figure 8, C_ENTR seems roughly constant during the nighttime but varies across the different classes of simulations. I would hypothesize that the differences between C_ENTR across the simulations depends on things like the jumps across the inversion, free tropospheric humidity, decoupling and perhaps other quantities as well, so the differences in the outcomes, e.g., positive/negative d(LWPc)/dt, might be able to explained in terms of those quantities. As noted above, this is pretty speculative, but (if it works) could potentially make the storytelling a bit cleaner.

This idea should work (with some modifications) for well-mixed BL with overcast clouds. Under this condition, the ENTR, RAD, and SURF terms for the $LWP_c$ budget are linear combinations of some $\theta_l$ and $q_t$ fluxes that are distributed over the $z_i$. (See Eq. 19 in the manuscript.) Further, the entrainment fluxes can be written as the products of the entrainment velocity ($w_e$) and $\theta_l$ and $q_t$ jumps across the inversion base ($\Delta\theta_l$ and $\Delta q_t$). Meanwhile, $w_e$ can be related to mean buoyancy flux in the BL (Eqs. 5–8 in Dal Gesso et al. 2014), which is again a linear combination of the $\theta_l$ and $q_t$ fluxes from various processes but with coefficients depending on normalized cloud depth (cloud depth divided by $z_i$). (See Eqs. 9–15 in Dal Gesso et al. 2014.) So, we should be able to write *each and the sum* of the ENTR, RAD, and SURF terms as fairly complicated linear combinations of four terms, that is, $\theta_l$ and $q_t$ fluxes by RAD and SURF.

For the CV budget, the ENTR and RAD fluxes have different interpretations from those for the BL budget. Consider ENTR as an example. Imagine a pocket of FT air gets engulfed into the BL and forms a downdraft that reaches the surface. In the CV budget, the ENTR term still assumes all this pocket of FT air warms and drys the CV. So, the flux due to the downdraft that leaves the CV from its bottom gets lumped into the BASE-n-LAT term. Similar for the RAD term. So, it is probably more complicated to use a linear combination of the RAD and BASE-n-LAT fluxes to represent the fluxes that drive the turbulence and to connect to $w_e$. Another complication is the connection between the LAT term to the TKE that can be used for entrainment.

To summarize, this is a very interesting idea that is worth more careful derivation and diagnosis in the future. For now, we show the C_ENTR term as suggested by the reviewer in Figure R1. We do see that during the night, C_ENTR is similar for three categories. During the day, C_ENTR for hi$\mathcal{D}$hiL increases dramatically until about 10:00 LT then decreases quickly to below 0.8, the average C_ENTR for lo$\mathcal{D}$loL and hi$\mathcal{D}$loL during the night time. Note that 10:00 LT is close to the time when the clouds in hi$\mathcal{D}$hiL start to break up. (See Figure 3b in the revised manuscript.) So, the quick increase in C_ENTR between sunrise and 10:00 LT is not affected by the LAT term. One possible explanation is that because clouds in the hi$\mathcal{D}$hiL category start with high $LWP_c$, the buoyancy flux from latent heat release in the cloud layer is much more important than the flux from the surface. Meanwhile, the weakening of the radiative cooling is less effective in reducing the buoyancy flux because the coefficient for $\theta_l$ flux in cloud layer is small. (See $\beta_{s,m}$ in Eq. 8 in Stevens et al. 2002.) Arguably, C_ENTR for hi$\mathcal{D}$loL follows similar evolution (C_ENTR peaks around 09:00 LT, about the same time as the clouds in this category start to break up) but with much weaker amplitude.

===========================

1/3: My understanding is that the "cloud liquid water path (LWPc)" refers to the mean cloud liquid water path in cloudy columns. If so, could this be stated explicitly somewhere, even if it is broadly understood by others in the field. Also, please explain somewhere how a "cloudy column" is defined.

We now add the definition of the "cloudy columns", that is, model columns with cloud optical depths greater than 1, to the abstract (L4 in the revised manuscript). Previously, this was not stated until L119 to L121 in the original manuscript. We have also slightly reworded the text there to make the definition more explicit. Please see L139–L140 in the revised manuscript.

4/94: The broad range of initial boundary layer temperatures (10K) suggests that the initial boundary layer is relaxing towards quasi-equilibrium with the SST throughout the day, as suggested by the surface temperature jump in figure 4c. It's great to have such a variety of cases here, but I found myself wondering how such a transients might impact the results. Note also, that the mesoscale organization will also be developing through the nighttime and that this transient might have some impact on the results (e.g., the timing of precipitation development overnight).

We notice the relaxing towards the SST in many cases, like the reviewer suggested.

To address this issue, we repeat all simulations with the SST set to 0.5 K warmer than the initial surface air temperature *for each case*. We find that the main features of both daytime and nighttime evolution are not affected by this change in configurations. Please see Section 6.2 in the revised manuscript.

We agree that the mesoscale organization also takes time to develop. This process could take days and is beyond what we can afford to simulate for an ensemble. We now note this in the text (L613–L615 in the revised manuscript). One of the co-authors of this paper is working on a paper on the mesoscale organization in this ensemble.

Since we focus on non-precipitating cases here, the impacts of this transient on precipitation development should be minimal.

10/277-278: The last subsection seemed to be foreshadowing a cloud-volume budget, so it's worth expanding BL and CV here for clarity.

The CV budget is the focus of the paper. We use the BL budget in Section 4.1 to diagnose the ENTR terms for the CV budget. We present the BL budgets in Section 4.2 because BL budgets are probably more familiar to readers and serve as a reference for the CV budgets.

We have slightly reworded the beginning of Section 4.2 to hopefully improve the flow. Please see L320–L322 in the revised manuscript.

Fig 6/bottom of page 10 (OPTIONAL): Given the degree of decoupling discussed previously, it's not surprising that the BL-budget predictions of LWPc are poor. If the authors agree, perhaps figure 6 could be removed from the paper, but a dashed or dash-dotted line could be added to the right column of figure 8 showing the d(LWPc)/dt prediction based on the BL-mean budget. That would make clear that the prediction is especially poor during the daytime and could be emphasized around the discussion of figure 8. However, if the authors believe that making this point clearly is

important for the reader, please show the prediction of d(LWPc)/dt explicitly in the left and right panels of figure 6. Showing the actual and residual does not have the same impact as showing the actual prediction (for me at least).

Fig 8: My understanding is that the budgets in figures five and seven are closed by definition so that the sum of the process tendencies and the actual tendencies are identical. Since the LWPc budgets are not necessarily closed, it would be useful to plot the predicted tendency based on the sum of the individual processes instead of (or alongside) the actual tendency. If figure 6 is removed, both the BL-budget and CV-budget predictions of d(LWPc)/dt could be included in the right column of figure 8.

Regarding the two comments on Figures 6 and 8:

We present the residual (RES) instead of the sum of the recovered LWPc tendency because RES is a term that we can group with other terms. This became important for the right column in Figures 8 and 13f in the original manuscript (Figures 7 and 12f in the revised manuscript).

For readers who are more used to the comparison between actual and predicted $dLWP_c/dt$, we add dashed lines to the right columns of both Figure 6 and Figure 8 (Figures 5 and 7 in the revised manuscript) to show the $dLWP_c/dt$ predicted by the sum of four main terms (ENTR, RAD, SUBS, and SURF for the BL budget and BASE-n-LAT for the CV budget).

We choose to keep Figure 6 (Figure 5 in the revised manuscript) but update the text to sharpen the message regarding the LWP$_c$ budget based on BL $\langle\theta_l\rangle$ and $\langle q_t\rangle$ budgets. We also introduce the decomposition of ENTR and SUBS and the combined term associated with $dz_i/dt$ here.

Note that Figures

11/304: I don't understand what is meant by "The warming strengthens the stratification of the sub cloud layer". Is it a change in the surface temperature jump? Is the base of the cloud volume high enough that "sub cloud" includes the transition layer and part of the cloud layer? Does the sub cloud layer actually depart from being well-mixed?

What we mean is, the radiative heating rate profiles show that the RAD term became more positive (warming) with height in the subcloud layer. This effect would strengthen the stratification of the sub cloud layer. See Figure S6 in the supplementary material. We also add text to point readers to Figure S6. See L348–L349 in the revised manuscript.

For deeper cases, the cloud bases are high enough to include surface-based mixed-layers and stratified layers. Please see Figure S1 in the supplementary material for profiles from an example case.

14/401-407: Does decoupling explain part of the correlation of LWPc velocity and z_i, since deeper boundary layers are likely more decoupled?

We agree that this could be a mechanism for the correlation we see in this dataset. We have added this to L440–L441 in the revised manuscript.

14/z_i scaling: It's interesting that the z_i scaling works so well, but I found myself wishing it had been more clearly motivated. Why do we have so much faith in the z_i scaling of these budget terms when the BL-budgets did so poorly in predicting d(LWPc)/dt? As an aside, the suggestion

above for computing C_ENTR resulted from efforts to understand the relationships between these budget terms better and seeking an alternative to the z_i scaling.

What we present here are the results during the nighttime, when the BLs are closer to well-mixed. Therefore, it makes sense to try the scaling by $z_i$.

16/section 6.1: See major comment 1a above.

Please see the response to Major Comment 1a.

18/538-539: Glassmeier et al (2021, https://doi.org/10.1126/science.abd3980) seem to argue that long timescales are important for LWP adjustments. In such circumstances, the surface flux adjustment might play a role in modifying the steady-state LWP relative to one computed using fixed/prescribed surface fluxes. If it's feasible, the authors could make an estimate of how the inclusion of interactive surface fluxes might have impacted the slope of the predicted d(log LWP)/d(log N) in Glassmeier et al.

A separate manuscript is being prepared to further the understanding of the timescale aspect of the LWP adjustment. It will be submitted soon. For the current work, we follow the suggestions by Reviewer #2 to remove the discussions on surface flux in the original Section 6.2. We summarize the results from 0.5 K warmer SST in the new Section 6.2.

===========================

Typographical/rephrasing suggestions (OPTIONAL):

14/404: "... sufficiently high to suppress _cloud base_ precipitation ..."

Thanks. We now add "cloud-base" before "precipitation" to be more specific. Please see L447 in the revised manuscript.

https://doi.org/10.5194/egusphere-2024-1033-RC2

General comment:

This paper investigates the diurnal behavior of an ensemble of stratocumulus-topped-boundary-layers produced by LES. While the topic is interesting, I find this paper somewhat confusing. In particular, I think it would be beneficial if the authors clarified the main message of the paper, as it currently seems too broad, and if they limited their numerous analyses to the most essential ones.

As this type of investigations definitely helps in broadening our knowledge on the topic, it should be explained early in the paper that the simulated cases are not likely to be observed in nature, especially due the assumption of zero wind, additional non-physical modifications of the surface fluxes calculation, and the assumption of fixed subsidence with varying inversion strength. In my opinion, the most valuable part is the LWP budget, although there seems to be an error in its derivation in addition to some confusing statements. My recommendation is major revision.

My general suggestions are:
- Simplify the message
- Remove the hysteresis plots and discussion
- Review equations for correctness
- Clarify the setup and simplifying assumptions
- Remove surface fluxes analysis

Thanks for these general comments. One reason why the paper seems too broad is that it needs to describe many aspects of this dataset to provide the background for other works using this dataset (Zhang et al. 2024 and one other manuscript that is in preparation). We have sharpened the messages of the current manuscript following the reviewers' suggestions, as discussed below.

Specific comments:

L2: Clarify how you construct your ensemble (what do you vary?). Provide a reason why it is large. You mention 3 categories first, but then only 2 important categories are discussed: coupled and decoupled layers, which is confusing. It would be helpful to mention that you only look at the statistics.

We have updated the abstract to indicate that the ensemble was constructed by perturbing the initial thermodynamic profiles and the initial aerosol conditions. (Please see L2–L3 in the revised manuscript.) Details on the 6 parameters that we perturb to change the initial profiles were described from L85 to L99 in the original manuscript (L93–L108 in the revised manuscript).

This LES ensemble covers a relatively broad range of conditions defined by these 6 parameters. The absolute number of the ensemble members is also relatively large as the community moves away from a few "golden cases" to exploring a wider range of environmental conditions. Instead of saying "a large ensemble of large-eddy simulations", we are now specific in the abstract by using "an ensemble of 244 large-eddy simulations". (The number of cases decreases from 245 in the original manuscript because we follow the reviewer's suggestion to further exclude one case whose max cloud top reached 1.9 km but not 2.0 km. Please see below.)

We have removed the "three categories" in the abstract. We do discuss all three categories in the main text.

We mostly report the statistics but also present the evolution of all or subsets of cases, especially in Section 5 (Figures 9 to 13 in the original manuscript or Figures 8 to 12 in the revised manuscript).

L5, L6: What do you mean by more coupled and decoupled clouds? Is coupling / decoupling a feature of clouds here?

Sorry for the confusion. We have updated the abstract to clarify that we are referring to the coupling state of the BL. Please see L4–L5 in the revised manuscript.

It would be important to show, at least in one figure, the most representative examples of the 'coupled' and 'decoupled' STBLs in terms of their vertical structure (mean profiles Thetal and qt profiles, qc, turbulence). It would also help link your analysis with observations more closely.

Thanks for this suggestion. We have added profiles for two cases, one more coupled and one more decoupled, in the supplementary materials. Please see Figure S1. We also add some description of these two cases to L171–L176 in the revised manuscript.

L10: "The time rate of change in the LWPc is more likely to be negative for higher LWPc and greater inversion base height (zi)." Unclear. Does it suggest that LWP more likely decreases at night for larger LWP?

The reviewer's interpretation is correct. We have updated the sentence to be concise and clear. Please see L10 in the revised manuscript.

L12: The sentence about 10h and 15% offset is difficult to understand and seems insignificant. Consider removing it.

We have removed this sentence from the abstract and the original Section 6.2 from the main text. However, testing the sensitivity of cloud behavior to the lower boundary condition is one objective of the current study. We have replaced Section 6.2 with results from an LES ensemble with SST set to 0.5 K warmer than the initial surface air temperature. This new subsection also addresses some concerns from Reviewer #1.

L32: Your 'early works' include papers that are more recent than 'recent works'. Please correct.

Thanks for pointing this out. We have changed "early work focused primarily on" to "Many previous works focused on". Please see L31–L32 in the revised manuscript.

L37: Explain that these are highly idealized simulations (e.g., with fixed boundary conditions like surface fluxes, for quasi-steady states, etc.). Is it only for different initial conditions? We need to keep a clear distinction between the real world and idealized and simplified simulations.

We agree that it is important to distinguish between real cases and idealized simulations. We have updated the text in the introduction to state that the simulations used in Glassmeier et al. (2019) were idealized in terms of fixed surface fluxes and so on. (Hoffmann et al. 2020 and Glassmeier et al. 2021 used the same simulations as Glassmeier et al. 2019.) (See L43–44 in the revised manuscript.) We have also updated the text to briefly introduce the configurations for Hoffmann et al. (2023). (See L57–58 in the revised manuscript.)

Here we provide an overview regarding the realism of the ensembles we have been working on in our group so far: one ensemble for Glassmeier et al. (2016), Hoffmann et al. (2020), and Glassmeier et al. (2021), another one used by Hoffmann et al. (2023) and Feingold et al. (2022), and a third one for the current manuscript and Zhang et al. (2024). All these ensembles fall into the category of idealized simulations. In each ensemble, the ensemble members differ only in initial thermodynamic profiles and initial aerosol conditions. All ensembles are configured with the same subsidence profile (i.e., divergence set to $3.75 \times 10^{-6}$ s$^{-1}$, following DYCOMS-II RF02), fixed SST, and 0 horizontal mean wind profile.

The representativeness of the base thermodynamic profiles and the treatment of the surface fluxes have been progressively improved:

The first ensemble centered around the DYCOMS-II RF02 thermodynamic profiles and lower boundary conditions (prescribed SST and surface fluxes). The ensemble members were generated by randomly specifying fix parameters (five for perturbing the initial thermodynamic profiles and one for initial aerosol conditions).

The second ensemble centered around the climatological thermodynamic profiles and SST from ERA5 (based on all months). The climatological thermodynamic profiles which were again perturbed to generate the initial profiles for ensemble members. This set also started to use interactive surface fluxes. However, there was still no horizontal mean wind. As a result, the surface fluxes were produced from local horizontal air velocity in the lowest level of the model domain and were weaker than RF02 surface fluxes but within the range of previous measurements and simulations of stratocumulus. (See the end of the second paragraph in Section 2 in Hoffmann et al. 2023.)

In the current ensemble, the ERA5 climatological thermodynamic profiles and SST were based on April, May, and June, as described in the Appendix. The $q_t$ jump was allowed to be weaker to have ensemble members with more humid FT. Also, a 7 m s$^{-1}$ horizontal wind speed was added to the calculation of the surface sensible and latent heat fluxes. The values of the resulting fluxes are closer to the values prescribed in RF02. In the current ensemble, we also added the diurnal cycle to make the simulations more relevant to the real world.

We believe that there are two major limitations that should be improved next. First, we need to perturb other cloud controlling factors (CCFs), e.g., the subsidence profile and the SST. Second, we need to capture the realistic co-variability between CCFs, like the correlation between inversion strength and subsidence (as suggested by the reviewer), SST and BL depth, and FT $\theta_l$ and $q_t$. To partially address these limitations, we re-ran all simulations for the current work with SST set to 0.5 K warmer than the initial surface air temperature *for each case* as a sensitivity run.

Hopefully our response here clarifies a few issues regarding the experiment design and its limitations for the three ensemble sets we have been using in our group. There are a few more comments below that are related to the experiment design and its limitations. We will provide point-by-point responses to them. We have updated the text correspondingly. Please see the point-by-point responses for the changes we make.

L43: More important than the number of simulations is that they covered a multi-dimensional space of idealized atmospheric conditions, aiming to represent – but only to some extent - the

observed variability of STBLs. Please clarify if that's the case.

It is indeed the case that we try to cover the multi-dimensional space of idealized atmospheric conditions, although a few other factors need to be improved in future work for better realism. We indicate in the revised manuscript that the six parameters perturbed to generate the ensmeble members were drawn independent of each other (L42 in the revised manuscript) and connect our results to observations to show that they capture some of observed features of STBLs (L45, L65, and L196–204 in the revised manuscript.)

Please explain if this dataset has been validated for the realism of the states it produces.

We now state in the introduction that the range of LWP from our LES ensemble is quite reasonable (L45 in the revised manuscript). We stated in Section 3.1 of the original manuscript that the surface fluxes reasonably cover a realistic range (L206–L208 in the revised manuscript). We believe that the main limitation that should be improved in future work is the realism of the co-variability between cloud controlling factors, no matter whether they are currently perturbed or not, which we emphasize at the end of the Summary (L608–L613 in the revised manuscript).

L53: When citing Hoffmann et al. (2023), please specify how idealized their simulations were. This is essential for understanding the strengths and limitations of these analyses. Additionally, please mention Chung et al. (2012), where they derived and proved via simulations the asymptotic values of cloud fraction for a range of steady states driven by various environmental conditions for stratocumulus-to-cumulus transitions.

We now state in the text that the main difference between Hoffmann et al. (2023) and previous work from our group is that it started to use ERA5 climatology and interactive surface fluxes (see L57–58 in the revised manuscript). We also state that both configurations (the ERA5 climatology and interactive surface fluxes) can be further improved, which is done in the current work. (See L61–L65 in the revised manuscript.)

We now cite Chung et al. (2012) at the beginning of Section 6.2 as a method to capture realistic co-variability in environmental conditions, before we present results showing the sensitivity to an alternative SST configurations. Please see L562–L565 in the revised manuscript.

L74: If it is only similar to Deardorff, what modifications does it include? In particular, how does it calculate horizontal vs vertical turbulent mixing coefficients?

The SGS has been documented in Khairoutdinov and Randall, 2003. "Similar to Deardorff (1980)" is the wording in Khairoutdinov and Randall (2003) when the developers of SAM described the SGS model. We have updated the text to cite both Khairoutdinov and Randall (2003) and Deardorff (1980) for clarity.

L79: Does this sentence cast doubt on the results from the two papers? Can you show the differences between the two approaches using a simple example (e.g., a 4-hour DYCOMS-II simulation)?

We update the code to predict total mass and number to take advantage of a well-acknowledged advantage of this method: it conserves total mass and number after applying a non-linear advection scheme. This is discussed in, e.g., Morrison et al. 2016MWR and Ovtchinnikov and Easter 2009MWR.

We have not noticed any major issues that invalidate the results from previous works. We do notice one distinct difference between the simulations produced with the old and new code: there is an artificial peak in the aerosol number concentration profile right above the inversion base with the old code (using the aerosol number concentration as a prognostic variable) but not with the new code (using the total number concentration as a prognostic variable). Please see Figure R2.

L84: What 'details' do you mean here? Do they refer to the differences between the two approaches, or the way the hydrometeors are calculated?

Sorry for the confusion. By "details", we are referring to a detailed discussion of the pros and cons of the new method.

We have updated the text to be more specific about this method. Please see L89–L92 in the revised manuscript.

L92: If the initial wind speed is 0 m/s, how realistic are these simulations? It should be highlighted in the abstract that this is an ensemble of no-mean-flow simulations, differentiating them from what we typically observe in the subtropics.

L102: I find this paragraph a bit confusing. The mean flow is 0 m/s, but you add 7 m/s to calculate surface fluxes. Why 7 m/s and not another value? While it is encouraging that you have put a lot of effort into making your environmental conditions consistent with observations, this surface condition, combined with the no-wind condition, appears highly unrealistic. In reality, a non-zero wind is needed to form a logarithmic wind profile, which is a significant source of TKE. You seem to focus on the limit of free convection where the mean flow approaches zero but still need to mimic reasonable surface fluxes. This is a very specific idealization of your setup that needs to be explained in the abstract and introduction, as such cases can never be observed.

Regarding the two comments on L92 and L102: The reviewer's assessment is correct that our setup is closer to free convection, meaning that there is no TKE generation from the vertical shear of the mean horizontal wind. We justify our choice of 7 m s$^{-1}$ wind speed that is added to the calculation of the surface latent and sensible heat fluxes with the following arguments. First, this value, coming from ERA5 statistics, is also similar to the surface wind speeds in Kazil et al. 2016 (black lines in their Figure 5a), which were produced from specifying ug/vg based on DYCOMS-II RF01. (We have added comments to Section 2 to clarify these points. See L114–L115 in the revised manuscript.) Second, as mentioned earlier, the surface fluxes from this wind speed cover a realistic range.

L108: You seem to use only one value of the large-scale divergence for the entire ensemble, and it matches that from DYCOMS-II RF01 (Stevens et al. 2005), which needs to be explained. That case was actually characterized by strong subsidence producing large T and q gradients at the top of STBL. Typically, weaker subsidence is associated with weaker inversion. Have you looked into the relationship between inversion strength and subsidence (cf. Wood and Bretherton 2006) to clarify it?

The reviewer touches upon two limitations of the current study that we now acknowledge in the introduction (L41–L43 in the revised manuscript), Section 3.2 (L200–L203 in the revised manuscript), and emphasized again in Section 6.2 (L558–L559 in the revised manuscript) and Summary (L612–L613 in the revised manuscript): Although the ensemble members span a wide range of environmental conditions, first, they do not perturb all cloud controlling factors (CCFs), e.g., the divergence, and second, they do not capture the realistic co-variability between CCFs, like the correlation between inversion strength and subsidence.

Your ERA5 climatology for SST is 292.4K, whereas Stevens et al. 2005 uses 292.5K. I can understand that difference, although I first thought you followed their study. However, your choice for the 7 m/s wind speed near the surface explained as ERA5 climatology matches one of the geostrophic wind components from Stevens et al. 2005. That is difficult to understand because its magnitude near the surface would likely be a small fraction of the geostrophic wind (Fig. C1 therein). ERA5 seems to be too coarse to provide a credible information about wind magnitude just above the surface.

We specify the $7 \, \mathrm{m \, s^{-1}}$ entirely based on ERA5 climatology. The similarity between this wind speed and ug component in DYCOMS-II RF01 is a coincidence.

We agree that the surface wind speed should be slower than the geostrophic wind speed. This is indeed the case in Kazil et al. 2016, where the ug/vg were prescribed following DYCOMS-II RF01 (ug = $7 \, \mathrm{m \, s^{-1}}$ and vg = $-5.5 \, \mathrm{m \, s^{-1}}$) but the surface wind speed is about $7 \, \mathrm{m \, s^{-1}}$ (see black lines in their Figure 5a).

L109: Explain the reason for using such non-uniform grids (dz/dx=1:20). A 200 m grid length is even more than what is typically used for LES of deep convection. I am quite surprised by this choice because Sc circulations are generally weak and very local. Assuming that effective resolution is typically around 4-6dx (Skamarock et al. 2014), you may not be able to represent the relevant small-scale dynamics here. What is the partitioning between explicit and subgrid TKE for this setup in the cloud and subcloud layers? Is it still LES or more of a gray-zone approach? In your free-convection limit it may give you 1km size of the smallest eddies. Did you look into that? I would be curious to see some examples of the flow patterns.

Previous works suggested that a 200-m grid spacing is not necessarily a poor choice. For example, Wang and Feingold 2009JAS Part 1 showed that the differences between closed- and open-cell clouds captured by 300-m horizontal resolution simulations are similar to those simulated with 100-m resolution. Pedersen et al. 2016 reported a better agreement between simulations with non-uniform grids (dz/dx = 1:7; dx = 35 m, 70 m, and 105 m) and DYCOMS-II RF01 observations than two sets of isotropic grids (10-m and 15-m), which they linked to the anisotropic turbulence in the inversion layer. Besides, given the limited computational resources, we also need to balance between using higher horizontal resolution and exploring a broader range of environmental conditions with many cases for the current work. With all these considerations, we choose to use a 200-m grid spacing. We have updated the text to explain this choice. Please see L121–L125 in the revised manuscript.

Wang and Feingold 2009JAS Part 1 did note that with a finer horizontal resolution, the BL is more likely to be well-mixed. We anticipate that the fraction of cases falling into three categories will differ if we increase the horiztonal resolution, but the behavior of each category will remain less affected.

As suggested by the reviewer, we also check the ratio between the resolved and total TKE for representative cases. More than 90% of total TKE is resolved. Please see Figure R3.

L110: If your initial well mixed layer can be as high as 1300m (L95), and your inversion can be very weak or even non-existent (L95) then one can expect the convective layer to get deeper than 2km during daytime. Is it justified to apply a damping layer above 2km? Can you please demonstrate that all the simulated cloud layers have their tops much below 2km? If some of them reach 1.9-2km then I think they should be excluded from the analysis.

Our BL $\theta_l$, $\theta_l$ jump, and inversion base height are specified independently of each other. In particular, the $\theta_l$ jump will always be at least 6 K for any initial profile (L94 in the original manuscript), no matter whether it passes the criteria that we describe from L96 to L99 in the original manuscript (L105–L108 in the revised manuscript).

The choice of the damping layer is independent of the clouds, but we do exclude cases with max cloud top height above 2 km. We appreciate the reviewer's suggestion to further lower this threshold to 1.9 km. It is reasonable considering the results in Moeng et al. 2005JAS that the inversion determined from scalar mixing is higher than the highest cloud top. Fortunately, there is only 1 case out of 245 cases whose max cloud top height reaches 1.9 km but not 2.0 km. The impacts of including this case on the results shown in the original manuscript should be minimal. For the revised manuscript and this response file, we have adopted 1.9 km as the new threshold.

L111: What day of the year is it? Are you following Stevens et al. 2005 for the length of day here?

The day is May 16. Since we use ERA5 climatology from April to June to configure the simulations, we pick the center of this three-month period. We have added this information to the description of the simulation configurations. Please see L129–L131 in the revised manuscript.

L113: Do your cases precipitate in general? What controls the autoconversion rate? If you only focus on non-precipitating cases, is it because you suppress autoconversion or you only select cases with shallow cloud layers?

We do have cases that precipitate but we focus on non-precipitating cases in current work, which we define as cases where the cloud base precipitating rate never exceeds 0.5 mm d$^{-1}$. (See L113 in the original manuscript or L132 in the revised manuscript.) We do not directly use a cloud depth criterion to choose cases with shallow cloud layers. The autoconversion is always turned on and calculated based on the bin-emulating approach of Feingold et al. (1998).

L115: I now understand that you actually exclude the cases where convection reaches 2 km. However, if your damping layer starts at 2 km, there is typically an entrainment interfacial layer right above the cloud layer (Haman et al. 2007) that is part of the circulation and needs to be accounted for. My suggestion would be to exclude the cases with cloud top height above 1.9 km or so.

Please see our previous response to the reviewer's earlier comments on L109.

L130: van der Dussen et al. (2013) used a simple flux-based measure of decoupling useful in finding temporary decoupling conditions between the subcloud and cloud layers. How does your index compare to theirs?

We calculated the simple flux-based measure of decoupling, $R_{q_t}$, following Eq. 13 in van der Dussen et al. (2013). Qualitatively, both $\mathcal{D}$ and $R_{q_t}$ suggest that the BL is more coupled during the nighttime and more decoupled during the daytime; both metrics suggest that the BLs in the lo$\mathcal{D}$loL are more coupled than the BLs in the other two categories during the daytime. Please see Figure R4.

Please put your results in the context of Nowak et al. (2021) on coupled and decoupled STBL.

We now make connection to Nowak et al. (2021) when we present the representative cases. Please see L172–176 in the revised manuscript.

L138: This is a typical diurnal evolution of Sc; see van der Dussen et al. (2013) or Smalley et al. (2024).

We add a paragraph to the end of Section 3.2 to connect the diurnal cycles of our ensemble members to observations. Please see L196–L204 in the revised manuscript.

L140 and Fig 2c: what is your LWP criterion for loDloL? This category significantly overlaps with loDhiL, which makes it unclear why they are considered as two different categories. My understanding is that analyzing your data in terms of coupled vs decoupled STBLs may help you reach a broader audience.

There is no LWP criterion for lo$\mathcal{D}$loL. The upper bound of LWP for cases in this category is 180 g m$^{-2}$, which we used as the LWP criterion to separate hi$\mathcal{D}$loL and hi$\mathcal{D}$hiL. Please see the definition for the hi$\mathcal{D}$loL category in L160–L161 in the revised manuscript.

We agree that there is some overlap between lo$\mathcal{D}$loL and hi$\mathcal{D}$loL in the plane of LWP$_c$ and $z_i$. This is not surprising given that $\mathcal{D}$ does not show distinct multiple peaks with a valley at the threshold that we choose ($\mathcal{D} = 1$). Please see Figure R5. In other words, our criterion only qualitatively separate coupled and decoupled groups.

We agree that the presence of both coupled and decoupled STBLs makes this ensemble more valuable. And the purpose of categorizing the cases by the relative decoupling index $\mathcal{D}$ is exactly to separate these two conditions.

L163: I don't think this is a hysteresis, it looks more like a fraction of an open loop. The size of the loop is (not surprisingly) the largest for the largest LWP values, as we typically observe. You seem to prefer analyzing the data in various parameter spaces, which sometimes poses questions on what is the purpose of it. It may be worth considering to just focus on several main pieces of the analysis and reduce complexity of your analysis. I suggest to remove this panel.

We agree that "hysteresis" is not the most appropriate term here. We have updated the text to remove it and describe the evolution as an open loop. Please see L188–189 in the revised manuscript.

We choose not to remove this panel but add a paragraph to the end of Section 3.2 to connect the diurnal cycles of our ensemble members to observations. Please see L196–L204 in the revised manuscript.

In fig 3, what is the envelope of the results for each of the categories? Please add some shading.

We now add shadings to the curves in Figures 3a and 3b to show the range of 10$^{th}$ to 90$^{th}$ percentiles.

Fig 3 c shows that all of the LWP curves basically collapse to similar values at the end of the day (see my comment to L138).

We agree with reviewer's observation. We noted in L159 the original manuscript (L185 in the revised manuscript) that this suggests that the diurnal cycle strongly constrains the range of LWP.

We now also add references to van der Dussen et al. (2013) regarding this behavior (L186 in the revised manuscript).

Both Fig. 1 and Fig 3 look at the evolution of LWP but in different parameter spaces. Fig 1. Clearly shows that N is not a control parameter as practically all the trajectories are vertical. Please consider simplifying the message (merging Figs 1 and 3?).

The purpose of Figure 1 is to provide a connection to previous work (Glassmeier et al. 2019, Hoffmann et al. 2020, and Glassmeier et al. 2021). Also, it provides an overview of all cases before we categorize cases and present the cloud behavior based on the composite in Figure 3. We now add a sentence to make it clear that $N_d$ does not change much when we describe Figure 1 (L144–L145 in the revised manuscript).

Fig 4. – I suggest to remove it or move it to supplemental material. This may help clarify the message you want to deliver, which I think is on the evolution of the cloud layer (LWP, cf).

We have moved this figure to Figure S2 in the supplementary material.

L194: Please clarify why you need to use LWPc in your detailed budget analysis? For example, van der Dussen focuses on the LWP tendency for adiabatic Sc layers.

Van der Dussen et al. 2013 focused on a few terms that are important for the BL *moisture budget*: (1) moisture flux at the surface and cloud base and (2) surface precipitating rate. The connection between these moisture budget terms and the *LWP budget* is more direct for the surface precipitation rate than for the moisture flux.

We believe that the reviewer was referring to van der Dussen et al. 2014, which focused on the LWP tendency for adiabatic Sc layers. Compared with that work, our work deals with partial cloudiness. As a result, the cloud *layer* is not well-mixed. By focusing on the LWP$_c$, we avoid mixing the cloudy part and clear-air part of a cloudy layer together.

In the original manuscript, we cited Van der Dussen et al. 2014 as one example using MLT-based budget analysis. We now make a more specific reference near the beginning of Section 4 to clarify the difference between our approach and theirs. Please see L241–L242 in the revised manuscript.

L201-205: Please clarify how your CV approach is "also based" on this observation. Is CV a cloudy fraction of the cloud layer?

The mixed-layer based budget only applies to a volume that is close to well-mixed. The fact that the cloud is entraining suggests that some turbulence is mixing the cloud volume.

Yes, a CV is the cloudy part of a cloud layer. Please see Figure S4 for an illustration.

Eq.5: Since you already introduced fc, I suggest to use one symbol for area fraction (fc vs f). Consider using z_b rather than z0 to distinguish from the surface. I think it should be Psi(t), M(t), etc.

We try to be general at the beginning of the derivation. $f$ is the volume area fraction. If the volume is a CV, then $f$ is the same as $f_c$. If the volume is the whole BL, then $f$ is 1. Same justification for using $z_0$. It is the volume base. For a CV, it is the same as $z_{cb}$; for a BL, it is 0. We have updated the text to clarify. Please see L251–L242 in the revised manuscript.

We now add $(t)$ to $\Phi$ and $M$ in Eqs. 5 and 6 and later state that "$(t)$" for most time-dependent variables is omitted to simplify the notation (L264 in the revised manuscript).

This is correct and we do use the temporally-varying values of $\langle \rho_0 \rangle$ in our calculations. What caused the confusion was that we did not include $(t)$ following $\langle \rho_0 \rangle$ to explictly indicate this time-dependence. We now (1) note that $\langle \rho_0 \rangle$ is a function of $t$ where $\langle \rho_0 \rangle$ first appears (Eq. 6 in the revised manuscript) and (2) state that "$(t)$" for most time-dependent variables are omitted to simplify notation (L264 in the revised manuscript).

L217: Is psi(z,t) the mean profile or is it the mean over CV only?

It is the mean profile for the volume of interest. So, for the BL budget, it is the domain mean; for the CV budget, it is the CV mean. We now clarify this in L259 in the revised manuscript.

L220: What derivation did you mean here? Do you follow the derivation for the budget from their paper? If you mean the decomposition, then citing Kazil et al. should take place around L.222.

Kazil et al. 2016 presented a budget analysis for the total moisture in the BL (i.e., "vertically integrated water mass per horizontal area"). We track both the total tracer ($\theta_l$ and $q_t$) amount and the total air mass in the volume of interest to get the $\langle \theta_l \rangle$ and $\langle q_t \rangle$ budget. The main benefit is that we can have exact ENTR terms for BL $\langle \theta_l \rangle$ and $\langle q_t \rangle$ without resorting to "jumps", which are not straightforward to estimate. We do not follow the exact decomposition in Kazil et al. 2016.

We now cite Kazil et al. 2016 in L252–254 in the revised manuscript and specifically refer to their BL total water budget.

Eq. 8: This equation doesn't seem correct. I think the 2nd term should read as -/M^2 dM/dt. Besides, since Psi=Psi(t,z), and M=M(t,z), I think these should be partial derivatives.

We have double checked our derivations per the reviewer's suggestion. We agree that the 2nd term in Eq. 8 could be written as

$$- \frac{\Phi}{M^2} \frac{\mathrm{d}M}{\mathrm{d}t}. \tag{1}$$

But with Eq. (7) in the original manuscript ($\langle \phi \rangle = \Phi/M$), it becomes

$$- \frac{\langle \phi \rangle}{M} \frac{\mathrm{d}M}{\mathrm{d}t}, \tag{2}$$

which is the form we used in Eq. (8). We have updated Eq. 8 in the revised manuscript to include this intermediate step to avoid confusion.

Regarding the second issue raised by the reviewer, both $\Phi$ and $M$ are integrated quantities for the volume of interest. They are functions of time, but not height. Please see Eqs. (5) and (6) in the original manuscript. So, we believe that it is correct to use $\mathrm{d}/\mathrm{d}t$.

Eq. 9 and next equations: they all carry the same mistake as Eq. 8. This puts in question all the analysis presented in this paper.

Please see our response to the previous comment.

Also, what do you mean by dM/dt|P? Why would different processes such as lateral entrainment or radiation change the mass of the volume that is fixed for given zo and zi (because rho0 is constant)? Wouldn't that imply dM/dt=0? Please clarify.

Apologies for the confusion. We write $\mathrm{d}M/\mathrm{d}t|_P$ as a general term for the budget. For radiation, exactly as the reviewer suggested, $\mathrm{d}M/\mathrm{d}t|_{\mathrm{RAD}}$ = 0. We noted in L235 in the original manuscript (L273 in the revised manuscript) that neither the RAD term nor the BASE term modifies $M$. The lateral entrainment, on the other hand, may be accompanied by changes in the cloud fraction. Thus, $\mathrm{d}M/\mathrm{d}t|_{\mathrm{LAT}}$ is not necessarily 0. Please see Eq. 18 in the original manuscript (or Eq. 18 in the revised manuscript).

For your mixed layer analysis, do you assume anything about your qt and thetal profiles in the cloud layer? Please list your assumptions for clarity.

To apply the MLT-based LWP budget analysis, we assume that both $q_{\mathrm{t}}$ and $\theta_{\mathrm{l}}$ are well-mixed in the volume. We have updated the text in L242 in the revised manuscript for clarity. Deviation from this assumption (for example, the cloud volume itself becomes stratified) results in large residual in the LWP budget.

Fig5-Fig8: Please add the envelopes of model spread to understand how different those tendencies are for your different subsets of cases.

Done. (Note that Figures 5–8 have become Figures 4–7 in the revised manuscript.)

L286: What do you mean by actual? Is it the real tendency from the model? Or is it the sum of all tendencies?

Yes, "actual" means the real tendency based on the time series of LWP$_{\mathrm{c}}$ from the model. We have updated the text to clarify (L330–L331 in the revised manuscript).

Fig. 11: This caption is confusing. "a few extra terms" doesn't sound precise.

We have updated this figure caption to be more precise, which now reads "Co-variability between ENTR, BASE-n-LAT, and SUBS terms for LWP$_{\mathrm{c}}$ budget and LWP$_{\mathrm{c}}$ at 04:40 LT". (Please note that Figure 11 have become Figure 10 in the revised manuscript.)

Section 6.2: You've already shown a lot in this paper and to me this part is not necessary.

We have replaced the original Section 6.2 with results showing the sensitivity to an alternative SST configuration. This also partially addresses some concerns from Reviewer #2 regarding the experiment design and some comments from Reviewer #1.

REFERENCES

Chung, D., G. Matheou, and J. Teixeira, 2012: Steady-State Large-Eddy Simulations to Study the Stratocumulus to Shallow Cumulus Cloud Transition. J. Atmos. Sci., 69, 3264–3276.

Haman, K.E., Malinowski, S.P., Kurowski, M.J., Gerber, H. and Brenguier, J.-L. (2007), Small scale mixing processes at the top of a marine stratocumulus—a case study. Q.J.R. Meteorol. Soc., 133: 213-226.

Nowak, J. L., Siebert, H., Szodry, K.-E., and Malinowski, S. P.: Coupled and decoupled stratocumulus-topped boundary layers: turbulence properties, Atmos. Chem. Phys., 21, 10965–10991, https://doi.org/10.5194/acp-21-10965-2021, 2021.

Skamarock, W. C., S. Park, J. B. Klemp, and C. Snyder, 2014: Atmospheric Kinetic Energy Spectra from Global High-Resolution Nonhydrostatic Simulations. J. Atmos. Sci., 71, 4369–4381.

Smalley, K. M., Lebsock, M. D., & Eastman, R. (2024). Diurnal patterns in the observed cloud liquid water path response to droplet number perturbations. Geophysical Research Letters, 51, e2023GL107323.

Stevens, B., and Coauthors, 2005: Evaluation of Large-Eddy Simulations via Observations of Nocturnal Marine Stratocumulus. Mon. Wea. Rev., 133, 1443–1462.

Wood, R., and C. S. Bretherton, 2006: On the Relationship between Stratiform Low Cloud Cover and Lower-Tropospheric Stability. J. Climate, 19, 6425–6432.

van der Dussen, J. J., S. R. de Roode, A. S. Ackerman, P. N. Blossey, C. S. Bretherton, M. J. Kurowski, A. P. Lock, R. A. J. Neggers, I. Sandu, and A. P. Siebesma (2013), The GASS/EUCLIPSE model intercomparison of the stratocumulus transition as observed during ASTEX: LES results, J. Adv. Model. Earth Syst., 5, 483–499.

**References**

Dal Gesso, S., Siebesma, A. P., de Roode, S. R., and van Wessem, J. M.: A mixed-layer model perspective on stratocumulus steady states in a perturbed climate, Quarterly Journal of the Royal Meteorological Society, 140, 2119–2131, https://doi.org/10.1002/qj.2282, 2014.

Deardorff, J. W.: Stratocumulus-capped mixed layers derived from a three-dimensional model, Boundary-Layer Meteorology, 18, 495–527, https://doi.org/10.1007/bf00119502, 1980.

Feingold, G., Walko, R. L., Stevens, B., and Cotton, W. R.: Simulations of marine stratocumulus using a new microphysical parameterization scheme, Atmospheric Research, 47–48, 505–528, https://doi.org/10.1016/S0169-8095(98)00058-1, 1998.

Feingold, G., Goren, T., and Yamaguchi, T.: Quantifying albedo susceptibility biases in shallow clouds, Atmospheric Chemistry and Physics, 22, 3303–3319, https://doi.org/10.5194/acp-22-3303-2022, 2022.

Glassmeier, F., Hoffmann, F., Johnson, J. S., Yamaguchi, T., Carslaw, K. S., and Feingold, G.: An emulator approach to stratocumulus susceptibility, Atmospheric Chemistry and Physics, 19, 10 191–10 203, https://doi.org/10.5194/acp-19-10191-2019, 2019.

Glassmeier, F., Hoffmann, F., Johnson, J. S., Yamaguchi, T., Carslaw, K. S., and Feingold, G.: Aerosol-cloud-climate cooling overestimated by ship-track data, Science, 371, 485–489, https://doi.org/10.1126/science.abd3980, 2021.

Hoffmann, F., Glassmeier, F., Yamaguchi, T., and Feingold, G.: Liquid water path steady states in stratocumulus: Insights from process-level emulation and mixed-layer theory, Journal of the Atmospheric Sciences, 77, 2203–2215, https://doi.org/10.1175/JAS-D-19-0241.1, 2020.

Hoffmann, F., Glassmeier, F., Yamaguchi, T., and Feingold, G.: On the roles of precipitation and entrainment in stratocumulus transitions between mesoscale states, Journal of the Atmospheric Sciences, 80, 2791–2803, https://doi.org/10.1175/JAS-D-22-0268.1, 2023.

Kazil, J., Feingold, G., and Yamaguchi, T.: Wind speed response of marine non-precipitating stratocumulus clouds over a diurnal cycle in cloud-system resolving simulations, Atmospheric Chemistry and Physics, 16, 5811–5839, https://doi.org/10.5194/acp-16-5811-2016, 2016.

Kazil, J., Yamaguchi, T., and Feingold, G.: Mesoscale organization, entrainment, and the properties of a closed-cell stratocumulus cloud, Journal of Advances in Modeling Earth Systems, 9, 2214–2229, https://doi.org/10.1002/2017MS001072, 2017.

Khairoutdinov, M. F. and Randall, D. A.: Cloud Resolving Modeling of the ARM Summer 1997 IOP: Model formula- tion, results, uncertainties, and sensitivities, Journal of the Atmospheric Sciences, 60, 607–625, https://doi.org/10.1175/1520-0469(2003)060<0607:CRMOTA>2.0.CO;2, 2003.

Moeng, C.-H., Stevens, B., and Sullivan, P. P.: Where is the interface of the stratocumulus-topped PBL?, Journal of the Atmospheric Sciences, 62, 2626–2631, https://doi.org/10.1175/JAS3470.1, 2005.

Morrison, H., Jensen, A. A., Harrington, J. Y., and Milbrandt, J. A.: Advection of coupled hydrometeor quantities in bulk cloud microphysics schemes, Monthly Weather Review, 144, 2809–2829,

https://doi.org/10.1175/MWR-D-15-0368.1, 2016.

Ovtchinnikov, M. and Easter, R. C.: Nonlinear advection algorithms applied to interrelated tracers: Errors and implications for modeling aerosol–cloud interactions, Monthly Weather Review, 137, 632–644, https://doi.org/10.1175/2008MWR2626.1, 2009.

Pedersen, J. G., Malinowski, S. P., and Grabowski, W. W.: Resolution and domain-size sensitivity in implicit large-eddy simulation of the stratocumulus-topped boundary layer, Journal of Advances in Modeling Earth Systems, 8, 885–903, https://doi.org/10.1002/2015MS000572, 2016.

Randall, D. A.: Stratocumulus cloud deepening through entrainment, Tellus A: Dynamic Meteorology and Oceanography, 36, 446, https://doi.org/10.3402/tellusa.v36i5.11646, 1984.

Stevens, B.: Entrainment in stratocumulus-topped mixed layers, Quarterly Journal of the Royal Meteorological Society, 128, 2663–2690, https://doi.org/10.1256/qj.01.202, 2002.

van der Dussen, J. J., de Roode, S. R., Ackerman, A. S., Blossey, P. N., Bretherton, C. S., Kurowski, M. J., Lock, A. P., Neggers, R. A. J., Sandu, I., and Siebesma, A. P.: The GASS/EUCLIPSE model intercomparison of the stratocumulus transition as observed during ASTEX: LES results, Journal of Advances in Modeling Earth Systems, 5, 483–499, https://doi.org/10.1002/jame.20033, 2013.

van der Dussen, J. J., de Roode, S. R., and Siebesma, A. P.: Factors controlling rapid stratocumulus cloud thinning, Journal of the Atmospheric Sciences, 71, 655–664, https://doi.org/10.1175/JAS-D-13-0114.1, 2014.

Wang, H. and Feingold, G.: Modeling mesoscale cellular structures and drizzle in marine stratocumulus. Part I: Impact of drizzle on the formation and evolution of open cells, Journal of the Atmospheric Sciences, 66, 3237–3256, https://doi.org/10.1175/2009JAS3022.1, 2009.

Yamaguchi, T. and Randall, D. A.: Cooling of entrained parcels in a large-eddy simulation, Journal of the Atmospheric Sciences, 69, 1118–1136, https://doi.org/10.1175/JAS-D-11-080.1, 2012.

Zhang, J., Chen, Y.-S., Yamaguchi, T., and Feingold, G.: Cloud water adjustments to aerosol perturbations are buffered by solar heating in non-precipitating marine stratocumuli, EGUsphere [preprint], https://doi.org/10.5194/egusphere-2024-1021, 2024.

Zhou, X. and Bretherton, C. S.: Simulation of mesoscale cellular convection in marine stratocumulus: 2. Nondrizzling conditions, Journal of Advances in Modeling Earth Systems, 11, 3–18, https://doi.org/10.1029/2018MS001448, 2019.

[Figure]

Figure R1: Ratio between the ENTR term and the sum of the RAD and BASE-n-LAT terms for LWP$_c$ budget.

[Figure]

Figure R2: Aerosol number concentration profiles at the same time from two simulations for a randomly picked case from the simulations used in Glassmeier et al. 2021. One simulation was performed using aerosol number concentration ($N_a$ or NA) as the prognostic variable and the other simulation using total number concentration (NT).

[Figure]

Figure R3: The ratio between resolved and total TKE for the two example cases shown in Figure S1.

[Figure]

Figure R4: Comparison of two metrics for coupling states: the relative decoupling index $\mathcal{D}$ (Kazil et al. 2017) and $R_{q_t}$ (van der Dussen et al. 2013).

[Figure]

Figure R5: Distribution of $\mathcal{D}$ at 09:40 LT.

---

## Author Response (AR2)

We thank the reviewers for their supportive comments.

In the rest of this file, we show the comments from Reviewers #1 and #2 in red and blue, respectively, and our response in black.

After addressing reviewers' comments, we list other minor edits we have made since the first revision.

**Reviewer #1**

Review of "Diurnal evolution of non-precipitating marine stratocumuli in an LES ensemble" by Chen, Zhang, Hoffman, Yamaguchi, Glassmeier, Zhou and Feingold, Manuscript egusphere-2024-1033, First Revision

I thank the authors for their careful consideration of and responses to the many reviewer comments. The new set of SST+0.5K simulations was an unexpected bonus and strengthens the paper, I think. I would recommend that the paper be accepted after the authors have a chance to consider the minor comments below.

Recommendation: Minor Revisions

===========================

Specific/minor comments (10/277 means p. 10, line 277, ALL LINE NUMBERS FROM TRACK CHANGES VERSION):

7/204-212: While the other reviewer and I had criticisms of the way the forcings and initial conditions for the ensemble were chosen, there is no need to be overly apologetic in describing the ensemble and what one might learn from it. The ambition of these ensembles is impressive, and there is a lot to be learned from them. I believe the sentences about the limitations of the ensemble design be phrased in a more positive manner, something like "While design of the ensemble did not account for the observed co-variation environmental conditions in simulation configurations, ..., the ensemble captures features of the observed diurnal cycles of marine stratocumulus, suggesting that analysis of the statical behavior of the ensemble could be valuable."

Thanks to the reviewer for understanding. This is exactly the point that we tried to convey: this LES ensemble does capture a range of very realistic diurnal cycles in observed clouds and thus could be valuable.

Please see the updated text in L216–L218 in the tracking-changes file.

19/sec. 6.2: The residuals in the LWPc cloud volume budget for the SST+0.5K hiD cases are stronger during the afternoon than in the fSST ensemble. Is this mainly related to the smaller afternoon cloud fractions seen in some simulations in the SST+0.5K ensemble? Or is there some other cause?

We believe that it is because hiD clouds are more decoupled from the surface in the SST+0.5K set, comparing Figure S8 and Figure 2 (same as the plots in the files uploaded for the first revision). This is consistent with the weaker surface fluxes with narrower differences between SST and surface air temperature (comparing Figures S9 and S2).

Very decoupled conditions are challenging to handle because the assumption of a well-mixed cloud volume is less likely to hold. Also, rapid ascending/descending of the cloud base may introduce larger uncertainly in the BM term.

Under more decoupled conditions, we agree that cloud fractions would be smaller in SST0.5K+ set.

Please see updated text in L583–584 in the tracking-changes file.

20/591: Regarding, "... we explore the impacts of diurnal cycles and free tropospheric humidity on the cloud system evolution of non-precipitating marine stratocumuli ..." This manuscript explores much more than just the impacts of the diurnal cycle and FT humidity, so I would suggest that the list be made more inclusive (and longer) or rephrased as something like "... impacts of diurnal cycles and varying atmospheric states on ...". I'm not sure "atmospheric states" is exactly right, but some catchall phrase like that might be more accurate than just "FT humidity".

We agree that the current scope of the manuscript is broader than just the impacts of diurnal cycles and FT humidity, which was the starting point of this study. We now reword this sentence to make it more consistent with the current scope. Please see updated text in L591–593 in the tracking-changes file.

The texts in Abstract and Introduction have also been updated accordingly. Please see L2–L3 and L71–73 in the tracking-changes file.

============================

Typographical/rephrasing suggestions (OPTIONAL):

5/129: "anisotropic"

Done. Please see L129 in the tracking-changes file.

7/206: comma before "while"

Done. Please see L211 in the tracking-changes file.

**Reviewer #2**

The authors addressed most of my major concerns satisfactorily. I only have two minor comments:

- the reason for disregarding the states with cloud tops reaching the nudging layer is more profound than that stated in the paper: it is to avoid interfering with the Entrainment Interfacial Layer (Haman et al. 2007, Kurowski et al. 2009) that is the cloud-free region where the cloud-top dynamics still occurs. If we modify EIL, we modify the cloud layer as well.

Thanks for being specific about the mechanism. We now update the text to elaborate the reason for excluding these cases. Please see L145–L147 in the tracking-changes file.

- the argument for choosing 200m grid spacing needs more clarifications. Matheou and Teixeira (2019) showed that grid convergence for Sc is typically achieved for grid spacings of several meters. The problem of grid convergence from the DNS and LES perspectives, and the benefits of hi-res simulations, was also elaborated in Mellado et al. (2018). For the 200-m grid spacing, a significant fraction of small-scale turbulence is not resolved. SGS transport can compensate for that to some degree, but it is unclear from the description provided in the text how the model does it for such coarse resolutions. Stevens et al. (2005) showed that SGS schemes tend to overestimate entrainment even for the resolutions much finer than the one applied in this paper.

We have incorporated these works into our discussions following the description of the resolution. Interestingly, Mellado et al. (2018) suggests that, if one has to use a coarse resolution, it is better to also use a large aspect ratio. Please see updated text in the paragraph from L123–136 in the tracking-changes file.

References:

Haman, K.E., Malinowski, S.P., Kurowski, M.J., Gerber, H. and Brenguier, J.-L. (2007), Small scale mixing processes at the top of a marine stratocumulus—a case study. Q.J.R. Meteorol. Soc., 133: 213-226.

Matheou, G., and J. Teixeira, 2019: Sensitivity to Physical and Numerical Aspects of Large-Eddy Simulation of Stratocumulus. Mon. Wea. Rev., 147, 2621–2639.

Mellado, J. P., Bretherton, C. S., Stevens, B., & Wyant, M. C. (2018). DNS and LES for simulating stratocumulus: Better together. Journal of Advances in Modeling Earth Systems, 10, 1421–1438.

Kurowski, M. J., P. Malinowski, S. and W. Grabowski, W. (2009), A numerical investigation of entrainment and transport within a stratocumulus-topped boundary layer. Q.J.R. Meteorol. Soc., 135: 77-92.

Other than the changes required to address the reviewers' comments, we now make the following edits.

- Change "amongst" to "among" in L17–18 in the tracking-changes file.
- Define "SGS" to be later used in the discussion of the grid spacing. See L83–84 in the tracking-changes file.
- Adding acknowledgments. See L663–672 in the tracking-changes file.